# Group B *Streptococcus* Cas9 variants provide insight into programmable gene repression and CRISPR-Cas transcriptional effects

Kathyayini P. Gopalakrishna[1,7], Gideon H. Hillebrand[2,7], Venkata H. Bhavana[1], Jordan L. Elder[3], Adonis D'Mello [4], Hervé Tettelin [4] & Thomas A. Hooven [1,5,6✉]

Group B *Streptococcus* (GBS; *S. agalactiae*) causes chorioamnionitis, neonatal sepsis, and can also cause disease in healthy or immunocompromised adults. GBS possesses a type II-A CRISPR-Cas9 system, which defends against foreign DNA within the bacterial cell. Several recent publications have shown that GBS Cas9 influences genome-wide transcription through a mechanism uncoupled from its function as a specific, RNA-programmable endonuclease. We examine GBS Cas9 effects on genome-wide transcription through generation of several isogenic variants with specific functional defects. We compare whole-genome RNA-seq from Δ*cas9* GBS with a full-length Cas9 gene deletion; *dcas9* defective in its ability to cleave DNA but still able to bind to frequently occurring protospacer adjacent motifs; and *scas9* that retains its catalytic domains but is unable to bind protospacer adjacent motifs. Comparing *scas9* GBS to the other variants, we identify nonspecific protospacer adjacent motif binding as a driver of genome-wide, Cas9 transcriptional effects in GBS. We also show that Cas9 transcriptional effects from nonspecific scanning tend to influence genes involved in bacterial defense and nucleotide or carbohydrate transport and metabolism. While genome-wide transcription effects are detectable by analysis of next-generation sequencing, they do not result in virulence changes in a mouse model of sepsis. We also demonstrate that catalytically inactive dCas9 expressed from the GBS chromosome can be used with a straightforward, plasmid-based, single guide RNA expression system to suppress transcription of specific GBS genes without potentially confounding off-target effects. We anticipate that this system will be useful for study of nonessential and essential gene roles in GBS physiology and pathogenesis.

[1] University of Pittsburgh School of Medicine, Department of Pediatrics, Pittsburgh, PA, USA. [2] University of Pittsburgh School of Medicine, Program in Microbiology and Immunology, Pittsburgh, PA, USA. [3] The Cleveland Clinic, Clinical Laboratory Services, Cleveland, OH, USA. [4] Institute for Genome Sciences, University of Maryland School of Medicine, Baltimore, MD, USA. [5] Richard King Mellon Institute for Pediatric Research, University of Pittsburgh Medical Center, Pittsburgh, PA, USA. [6] UPMC Children's Hospital of Pittsburgh, Pittsburgh, PA, USA. [7] These authors contributed equally: Kathyayini P. Gopalakrishna, Gideon H. Hillebrand. ✉email: thomas.hooven@chp.edu

*S*treptococcus agalactiae (group B *Streptococcus*; GBS) is a gram-positive pathobiont that colonizes the gastrointestinal and genitourinary tracts of up to 30% of healthy adults, among whom it rarely causes disease[1–4]. In late pregnancy and early infancy, however, GBS can assume invasive characteristics leading to ascending chorioamnionitis, preterm labor, stillbirth, or severe neonatal or infantile infections including bacteremia, pneumonia, meningitis, and septic arthritis[1,3,5–7].

GBS encodes a Cas9 endonuclease, which is a key effector of its type II-A clustered regularly interspaced short palindromic repeat (CRISPR)-Cas9 system. CRISPR-Cas9 defends bacterial cells against foreign DNA. RNA transcribed from targeting templates within the chromosomal CRISPR array (crRNA) partially anneals to trans-activating CRISPR RNA (tracrRNA) to form guide RNA (gRNA) that complexes with Cas9. In the presence of DNA complementary to the gRNA targeting sequence and adjoined by an appropriate protospacer adjacent motif (PAM), Cas9 generates a double-stranded DNA cleavage at the site[8,9].

Detailed characterization and extensive molecular reengineering of the orthologous *Streptococcus pyogenes* (group A *Streptococcus*; GAS) CRISPR-Cas9 system led to a revolution in biomedical research[10–14]. Many of the early demonstrations of CRISPR-Cas9's potential used eukaryotic experimental systems. However, CRISPR-Cas systems have also been adapted as tools for studying a range of bacterial phenomena. In bacteria, CRISPR-Cas-based approaches have been used for genome editing, molecular barcoding in long-term colonization experiments, tunable transcriptional control, and generation of knockdown libraries to perform genome-wide studies of gene function[15–27].

In GBS, comparative sequencing of CRISPR arrays in 351 strains demonstrated diversity and frequent rearrangement of targeting sequences, suggesting adaptation to a broad range of environments that may select for or against mobile genetic elements[28]. This same study showed that, as in GAS, the GBS Cas9 enzyme requires a NGG PAM next to the target recognition site at its 3' end[28]. Subsequent work has further investigated lineage specificity of GBS CRISPR arrays and their use as barcodes for precision subtyping[29,30].

A separate line of inquiry has suggested that the GBS CRISPR-Cas9 system has a direct role in genome-wide transcriptional regulation uncoupled from its activity as a bacterial "immune system" evolved to detect and cleave foreign DNA. Spencer et al. showed that deletion of the *cas9* gene in GBS strain COH1 led to widespread altered transcription across multiple growth phases and that this transcriptional dysregulation limited GBS vaginal colonization and invasive disease in murine experimental models[31]. Dong et al. showed that isogenic deletion of the CRISPR array in a piscine GBS isolate led to upregulation of 236 genes and downregulation of 77 genes, with resultant virulence attenuation in in vitro and in vivo models of colonization and infection[32]. A subsequent examination of CRISPR array deficient GBS demonstrated significant upregulation of capsule production in the mutant, with corresponding surface property changes[33].

The mechanism of genome-wide regulatory changes in GBS CRISPR-Cas9 mutants is not clear. One possibility is that CRISPR-Cas9 interacts with other global regulators—such as the CiaR or the CovR/S system—that affect genome-wide transcription[31,32]. Alternatively, GBS CRISPR-Cas9 may directly regulate a broad array of bacterial genes, including virulence factors that influence colonization and invasion, through noncanonical molecular mechanisms not yet understood.

In this study, we test the hypothesis that genome-wide transcriptional dysregulation in GBS CRISPR-Cas9 mutants results from disruption of nonspecific PAM sequence scanning by the wild type (WT) CRISPR-Cas9 system. We use targeted mutagenesis of the GBS *cas9* gene to eliminate Cas9 PAM scanning, DNA cleavage, or both. Through RNA-seq transcriptomic study of these variants, we show that the phenotypic effects of Cas9 deletion stem from loss of PAM scanning functionality.

We also examine the potential for using catalytically deactivated Cas9 (dCas9) as a tool for targeted repression of gene transcription in GBS. We demonstrate that GBS dCas9 behaves similarly as GAS dCas9 and can be programmed using a straightforward plasmid-based approach to downregulate transcription of specific chromosomal or extrachromosomal targets.

The major contributions of this work are to provide insights about the underlying mechanism of genome-wide transcriptional regulation by the GBS CRISPR-Cas system and a demonstration of how dCas9 can be used for rapid, targeted expression knockdown of essential and nonessential GBS genes located on the chromosome or a plasmid.

## Results

**Amino acid sequence comparisons between GAS and GBS Cas9 identify orthologous active sites.** Relationships between the amino acid sequence of GBS Cas9 endonuclease and its molecular functions can be deduced from detailed studies of its GAS ortholog, SpyCas9 (Supplementary Fig. 1).

As a first step in locating DNA complementary to a targeting gRNA molecule, SpyCas9 binds and scans chromosomal DNA, transiently attaching at NGG PAM sites followed by DNA strand separation[34]. This scanning function in SpyCas9 depends on protein-DNA interactions mediated by active sites Arg1333 and Arg1335, which interface with the GG moiety of the PAM. Replacement of these arginine residues with alanine eliminates the PAM scanning function of SpyCas9. Because PAM scanning is a prerequisite for subsequent DNA strand separation and attempted pairing between the gRNA and the DNA sequence next to the PAM, SpyCas9 R1333A/R1335A is nonfunctional and will not cleave target DNA even when provided with a complementary gRNA construct[35].

Following PAM binding and DNA strand separation, successful pairing of gRNA with a complementary DNA sequence results in catalysis of a double-stranded cleavage event performed by Cas9 N-terminus RuvC and C-terminus HNH domains[36]. In SpyCas9, introduction of active site missense mutations D10A and H840A render these domains nonfunctional such that, even after successful PAM binding and gRNA pairing, the target DNA remains intact[37].

Using these well-defined aspects of SpyCas9 active domains, we identified the equivalent active sites on GBS Cas9 responsible for PAM scanning (Arg1339 and Arg1341) and double-stranded DNA cleavage (Asp10 and His845).

**Generation of chromosomally expressed GBS dCas9 variants for CRISPR interference.** We used an established temperature selection/sucrose counterselection mutagenesis strategy to produce missense mutations D10A and H845A generating chromosome-expressed dCas9 incapable of causing double-stranded DNA cleavage (Fig. 1A). We created dCas9 variants in two GBS strain backgrounds: A909 (serotype Ia, sequence type 7) and CNCTC 10/84 (serotype V, sequence type 26).

Adapting published SpyCas9 techniques, we developed an expression shuttle plasmid, p3015b, that encodes a single guide RNA (sgRNA) template compatible with GBS Cas9 and its derivative mutants. The purpose of sgRNA is to use a single transcript to "program" Cas9 with a DNA target, instead of using separate tracrRNA and crRNA sequences that, in nature, dimerize after transcription from distinct loci (Fig. 1B).

We incorporated GBS tracrRNA and crRNA sequences into the sgRNA coding sequence to ensure proper folding and dimerization

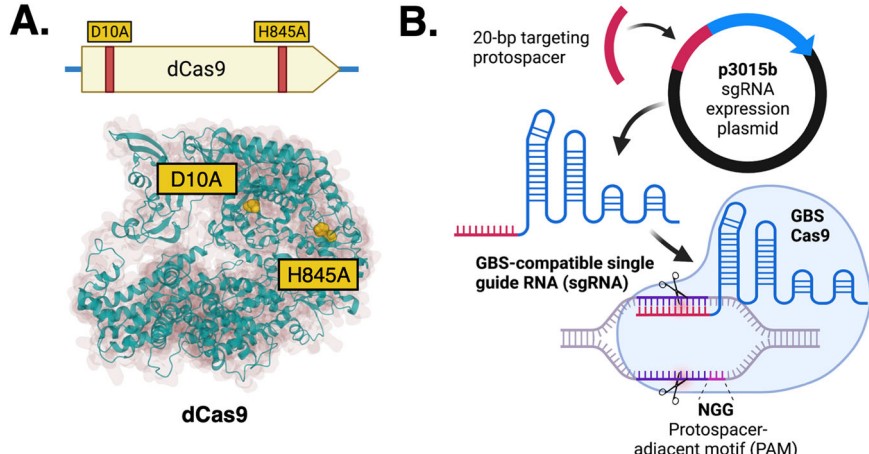

**Fig. 1 Molecular basis of GBS CRISPR interference.** We used allelic exchange mutagenesis approaches to generate catalytically inactive *dcas9* with D10A and H845A missense mutations (**A**). The p3015b sgRNA shuttle expression vector accepts a 20-bp targeting protospacer cassette sequence that, in GBS, produces a compatible sgRNA molecule that directs dCas9 to a complementary target sequence located adjacent to a NGG PAM site (**B**).

with GBS Cas9. The variable portion of the sgRNA coding sequence that directs Cas9 to a specific target exists as a 20-nt protospacer sequence flanked by BsaI restriction enzyme recognition sites. This arrangement allows straightforward excision and replacement of the protospacer using double-stranded, end-phosphorylated, custom-synthesized oligonucleotide cassettes (Supplementary Fig. 2)[25]. Protospacer annealing and subsequent plasmid propagation and miniprep were performed in *Escherichia coli* DH5α before transformation into electrocompetent GBS.

Because dCas9 can bind to a programmable genomic target without cleaving DNA at the site, it can be used to interrupt transcription initiation and propagation processes. This approach, called CRISPR interference (CRISPRi), has been applied broadly across numerous experimental systems including in some bacterial species[38] but is not widely used for GBS transcriptional control. Because CRISPRi enables rapid, flexible gene expression modulation—requiring only introduction of an appropriate targeting gRNA to direct dCas9 to downregulate transcription of a genomic site of interest—this strategy could be an alternative to laborious targeted GBS chromosomal mutagenesis.

**GBS CRISPRi using a luciferase reporter assay.** We tested whether our GBS dCas9 variants in CNCTC 10/84 and A909 could be directed to chromosomal or extrachromosomal genes to interfere with transcription. Foundational work with GAS *dcas9* demonstrated that CRISPRi transcriptional repression could be achieved with protospacers targeting either a gene's promoter or coding sequence but that CRISPRi efficacy is diminished by protospacers positioned near the 3' terminus or which target the sense strand instead of the antisense strand of the gene[37]. In fact, protospacers that target the sense strand of the coding sequence can lead to increased transcription over baseline[39].

First, we targeted a firefly luciferase (*ffluc*) reporter gene expressed from a plasmid, pFfluc. Maximum luminescence after normalization of a liquid culture and addition of the soluble Ffluc cofactor D-luciferin was compared in the presence of different p3015b targeting protospacers. The sites targeted along the *ffluc* promoter and coding sequence are shown in Fig. 2A. Both the sense/antisense strand effect and de-repression of 3' target sites were recapitulated in our GBS *ffluc* reporter gene targeting system. We normalized luminescence to the isogenic dCas9 strain bearing pFfluc and transformed with the non-targeting sham p3015b plasmid. Effective repression was achieved by targeting the sense strand of the promoter or the antisense strand of the coding

sequence, with diminished repression when the protospacer matched a site near the gene terminus. As in *E. coli*[39], targeting the sense strand of the *ffluc* gene led to increased expression over baseline in A909. In CNCTC 10/84, sense strand targeting did not affect expression significantly in either direction.

To assess the efficiency of the four possible NGG PAM sites in our GBS CRISPRi system, we identified an antisense *ffluc* GGG PAM at nucleotide position 316 whose corresponding CCC on the sense strand aligns with the open reading frame, encoding a proline residue. Codon degeneracy permits replacement of the third (3') C in the proline codon with any other nucleotide without changing the protein product.

We generated pFfluc isogenic variants where the only difference was the nucleotide at position 312, where we replaced the C with G, A, and T, producing CGG, TGG, and AGG antisense PAMs at position 316, respectively (Supplementary Fig. 3). We then conducted luminescence assays in our CRISPRi system with a protospacer targeting the sequence adjacent to the PAM at position 316. In both CNCTC 10/84 and A909, all four PAM variants resulted in effective luminescence repression (relative to isogenic controls in which sham targeted plasmid was introduced with the same pFfluc variant), but the AGG PAM was slightly less efficient than the other three variants (Fig. 2B).

**CRISPRi of GBS nonessential gene targets.** Next, we targeted two nonessential chromosomal genes with CRISPRi, assessing efficacy of gene repression through real time qPCR and phenotypic observation. Both CNCTC 10/84 and A909 encode the *cyl* operon, whose gene products biosynthesize the pigmented, hemolytic ornithine-rhamnopolyene cytotoxin β-hemolysin/cytolysin (βHC)[40–48].

The *cyl* operon is regulated by the CovR/S two-component system, a well-described regulator of loci distributed throughout the GBS genome. Many of the genes regulated by the CovR/CovS system have roles in virulence, including the *cyl* operon, which is repressed by CovR in its active state, marked by phosphorylation of residue D53, and upregulated when CovR is phosphorylated by the Stk1 serine-threonine kinase at residue T65[49–52]. Decreased *covR/S* expression in CNCTC 10/84 due to a single nucleotide polymorphism upstream of the two genes leads to relatively increased *cyl* expression and a hyperhemolytic, hyperpigmented phenotype[53]. We took advantage of the A909[high covR/low βHC] and CNCTC 10/84[low covR/high βHC] phenotypes diagrammed in Fig. 3A to test the predicted effects of CRISPRi on the two targets.

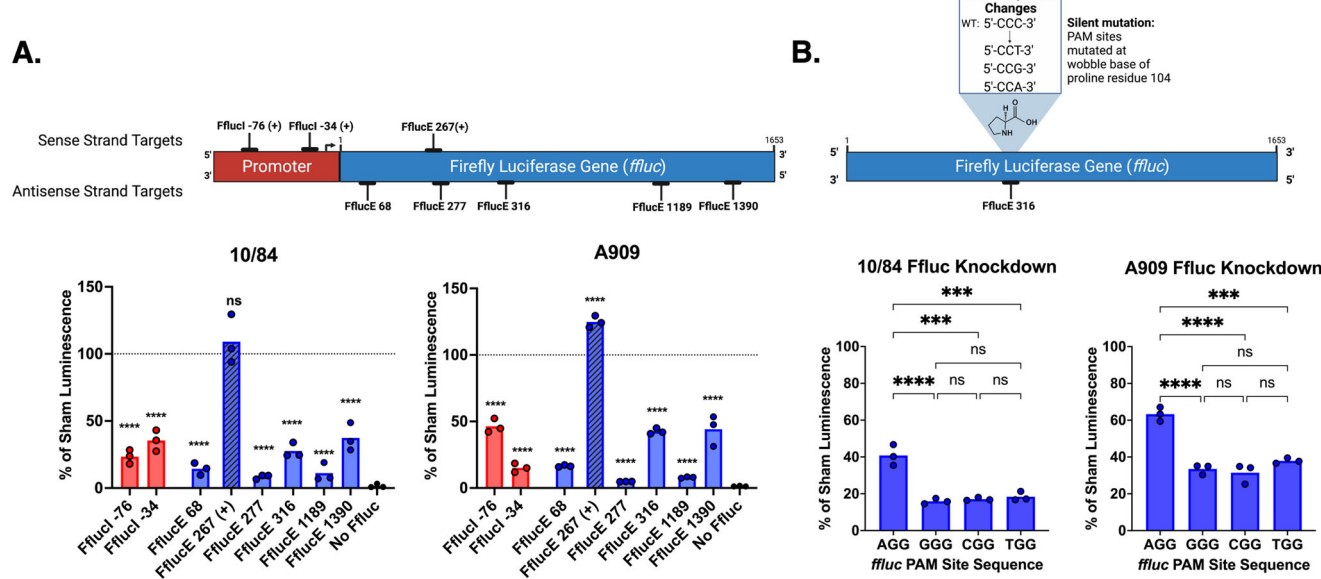

**Fig. 2 GBS CRISPR interference (CRISPRi) downregulates expression of a firefly luciferase (*ffluc*) reporter gene consistent with the previously established collision model of transcriptional blockade.** Multiple sites were targeted with complementary sgRNA protospacer sequences cloned into p3015b and transformed into CNCTC 10/84 and A909 *dcas9* strains, followed by standardized luminescence assays performed on the transformed strains. Targets were selected along the sense and antisense strands of DNA and were located in the promoter and along the coding sequence. Luminescence measurements revealed significant repression of Ffluc expression when the sgRNA-complementary target was on the antisense strand, with more repression caused by interference nearer the transcription start site than downstream (**A**, ****$p < 0.001$, one-way ANOVA with Bonferroni correction; comparison of each bar is to triplicate luminescence readings in the same strain transformed with the pFfluc plasmid and p3015b bearing a sham protospacer). In a set of point-mutated *ffluc* target genes in which all four possible PAM sequences were tested with the same sgRNA protospacer targeting nucleotide position 315 (see inset), expression knockdown was observed in all conditions, although significantly less downregulation was observed in both 10/84 and A909 when the target sequence was adjacent to the AGG PAM (**B**, ***$p < 0.005$ ****$p < 0.001$, one-way ANOVA with Bonferroni correction, ns = not significant). All data shown from $n = 3$ biological replicates with technical triplicates in each case; each data point is the mean of technical triplicates for that experiment.

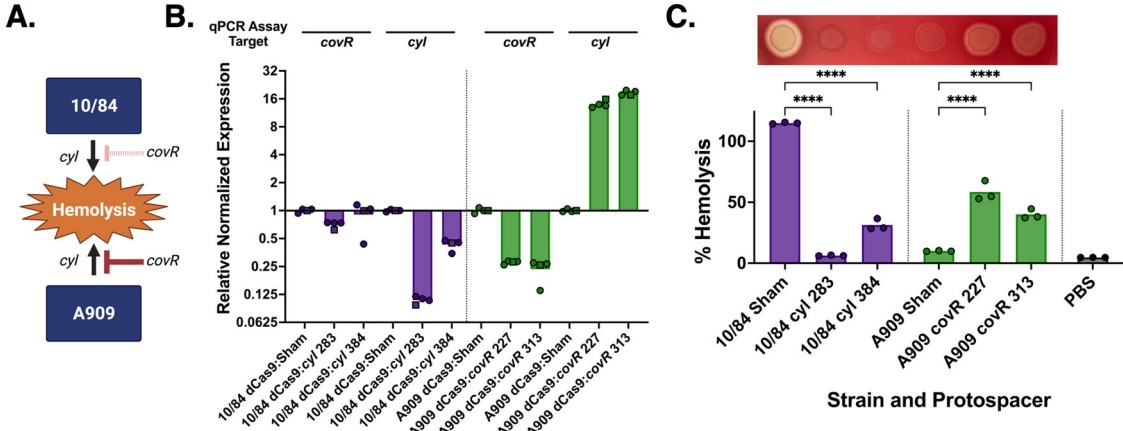

**Fig. 3 CRISPRi allows targeted knockdown of nonessential GBS genes.** CRISPRi against nonessential genes was demonstrated with the *cyl* operon, which generates the cytolytic toxin β-hemoylsin/cytolysin. In A909, the *cyl* operon is downregulated at baseline by CovR binding at a binding site upstream of the transcription start locus. In 10/84, the *covR* gene is minimally transcribed due to a promoter polymorphism (see ref. [53]), which renders it hyperhemolytic due to relatively increased *cyl* expression (**A**). CRISPRi targeting of *covR* and *cyl* in 10/84 and A909 *dcas9* strains led to expected repression or upregulation of genes by qRT-PCR (**B**, $n = 2$ biological replicates, indicated by data point shape; the second replicate was performed in technical triplicate, all data points shown). Confirmatory hemolysis assay results and plating on blood agar solid medium matched phenotype expectations from the regulatory model and qRT-PCR (**C**, ****$p < 0.001$, one-way ANOVA with Bonferroni correction, $n = 3$ biological replicates, each performed with technical triplicates; data points indicate technical triplicate means for each separate experiment).

As expected, CRISPRi targeting of *covR* and *cyl* led to decreased transcription in A909 and CNCTC 10/84, demonstrated by qRT-PCR measurement of cDNA from strains with two different targeting protospacers (Fig. 3B).

We then used these targeted and control strains in an in vitro hemolysis assay with washed human erythrocytes. We also plated samples of normalized liquid cultures on blood agar plates to visualize hemolytic activity from βHC expression. Figure 3C

shows the expected hemolytic phenotypes of our CRISPRi strains, with experimental targeting of the CNCTC 10/84 *cyl* operon resulting in decreased hemolysis and A909 *covR* targeting leading to increased hemolysis.

**CRISPRi of GBS essential gene targets**. Previous work used Tn-seq to characterize the genome-wide contributions of GBS genes to fitness[54–57]. These studies indicate that approximately 10–14% of GBS genes are essential for survival in rich media, with greater numbers of essential genes identified under conditions of environmental selection.

Determining the contributions of essential genes to a bacterial phenotype is challenging, generally requiring introduction of a chemical inhibitor[58] or a tunable method for downregulation[26,59,60]. CRISPRi offers an attractive approach to the latter strategy, potentially permitting partial repression of essential gene expression, enough to evaluate contributions to bacterial physiology without completely suppressing growth.

To assess whether our GBS CRISPRi system could be applied to suppression of an essential gene, we investigated down-regulation of carbon catabolite protein A (*ccpA*). CcpA is a central regulator of bacterial metabolism, optimizing energy production and utilization in conditions of diverse carbon sources[61–66]. CcpA has also been shown to influence the virulence potential of several pathogenic bacteria[61,63,65,67]. Our previous studies showed *ccpA* to be an essential gene in A909 when grown in tryptic soy broth[54]. A separate Tn-seq investigation of GBS strain CJB111 found *ccpA* to be essential in chemically defined media, but not Todd Hewitt broth[56]. This finding was reconfirmed by another group, which was able to generate a Δ*ccpA* deletion mutant in A909 grown in Todd Hewitt broth[68]. We also targeted the *gyrA* gene, which encodes the gyrase A subunit of the DNA gyrase enzyme, which is essential for bacterial viability[54,69,70].

We generated p3015b variants with targeting protospacers against three PAM sites on the antisense strand of *ccpA* and *gyrA*. The targeted regions are identical between CNCTC 10/84 and A909. Transformation of these plasmids into electrocompetent dCas9 GBS from these two strain backgrounds was successful for all six targeting protospacers, although colonies from transformation with the plasmid targeting *ccpA* gene position 160 were noticeably smaller and slower growing than transformations with plasmid targeting positions 861 or 886 (Supplementary Fig. 4). This observation was consistent with targets closer to the transcription initiation site showing more repressive effect than targets nearer to the stop codon.

This delayed growth phenotype was also observed in liquid culture (Fig. 4A, B). RT-qPCR analysis of mid-log *ccpA* expression in both strains aligned with the growth phenotypes, with more *ccpA* downregulation observed in the slow-growing strain with the position 160 target plasmid compared to the faster growing strains with targets at the 861 and 886 positions. The *gyrA* targeted dCas9 strains also showed growth defects, which were also influenced by targeting site position within the coding sequence (Supplementary Fig. 5).

**RNA-seq of CRISPRi targeted strains shows expected effects of gene repression without off-target epiphenomena**. We used transcriptomic analysis to examine the effects of GBS CRISPRi gene expression repression at genome-wide scale. We sought to assess whether specific locus targeting with dCas9 had tran-scriptomic effects limited to the intended target or, alternatively, if off-target epiphenomena were observed.

For these experiments, we returned to CNCTC 10/84 *dcas9* targeted to the *cyl* locus and A909 *dcas9* targeted to the *covR* transcription factor gene. If dCas9-mediated targeting in GBS were limited to the locus complementary to the 3015b-encoded sgRNA sequence—with minimal off-target effects—we would expect *cyl* targeting in CNCTC 10/84 *dcas9* to result in isolated knockdown of the *cyl* polycistronic operon, which does not have widespread influence on transcription of other genes. In A909 *dcas9* targeted to the *covR* gene, we would expect decreased expression of the *covR/covS* polycistronic gene pair. We would also expect specific expression changes in the CovR/CovS regulon, which is well-characterized, including in a recent, detailed study by Mazzuoli et al., in which the team used RNA-seq and CHiP-seq to formally define the set of genes directly regulated by the two-component system in GBS strain BM110[71].

Results in Fig. 5 (see Supplementary Data 1) show normalized RNA sequencing alignment coverage, genome-wide, in CNCTC 10/84 *dcas9* with a *cyl* operon sgRNA protospacer cloned into p3015b and A909 *dcas9* with a *covR*-targeting protospacer. In both strains, targeted transcriptomes were compared to isogenic controls, grown under the same conditions, with p3015b bearing a non-targeting control protospacer cassette, which we denote as the "sham" protospacer. Each targeted and non-targeted strain was grown in triplicate biological replicates, which were sequenced as independent RNA libraries. Data shown in Fig. 5 represent consensus, normalized alignments from all three replicates of each variant.

Comparing coverage between the targeted and non-targeted strains revealed the expected findings. In 10/84 *dcas9* targeted against the *cyl* operon, we observed a clear, isolated, drop in expression of the *cyl* genes starting upstream of nucleotide position 283 in the *cyl* operon, where the gRNA-complementary sequence is found, and returning to baseline expression at the end of the polycistronic transcript (Fig. 5A). This is consistent with the "CRISPRi collision model," in which dCas9 targeted to the antisense strand disrupts the RNA polymerase complex as it traverses the coding sequence, leading to reduced transcription of the downstream sequence[37].

In A909 *dcas9* targeted against *covR*, we observed reduced expression of the *covR/covS* transcript, as expected (Fig. 5B). Also, as expected, genome-wide transcriptional changes in the *covR* knockdown A909 variant aligned with predictions based on the defined BM110 regulon. Orthologous, regulated genes in A909 were repressed or upregulated in concordance with predictions from Mazzuoli et al., with no evidence of significant off-target influences (Fig. 5C, D). Incidentally, RNA-seq of the A909 dCas9 strain revealed auto-excision of a 37,412-bp prophage region located between nucleotide positions 552,394 and 589,805 (indicated by the asterisk in Fig. 5B), which was subsequently confirmed by whole-genome sequencing.

**Additional Cas9 variants show distinct phenotypes**. While several studies have shown that GBS Cas9 influences genome-wide transcription[31–33], the mechanism by which this influence occurs is not clear. We created and studied additional Cas9 variants to explore the hypothesis that nonspecific PAM scanning by wild type (WT) Cas9 leads to changes in transcription.

After the *dcas9* GBS strains, we generated *scas9* variants with an intended defect in PAM scanning due to missense mutations R1339A and R1341A. We also generated an allelic exchange Δ*cas9* deletion mutant (Fig. 6A, B). In the final step of GBS mutagenesis, the pMBsacB temperature- and sucrose-sensitive plasmid is excised from the chromosome and cured. The plasmid excision step can result in one of two outcomes: generation of the desired mutant or reversion to the original sequence. As controls for our mutant *scas9* and Δ*cas9* strains we retained isolates of the corresponding revertant strains. Because *dcas9* was generated by two separate mutagenesis steps (to establish the two missense

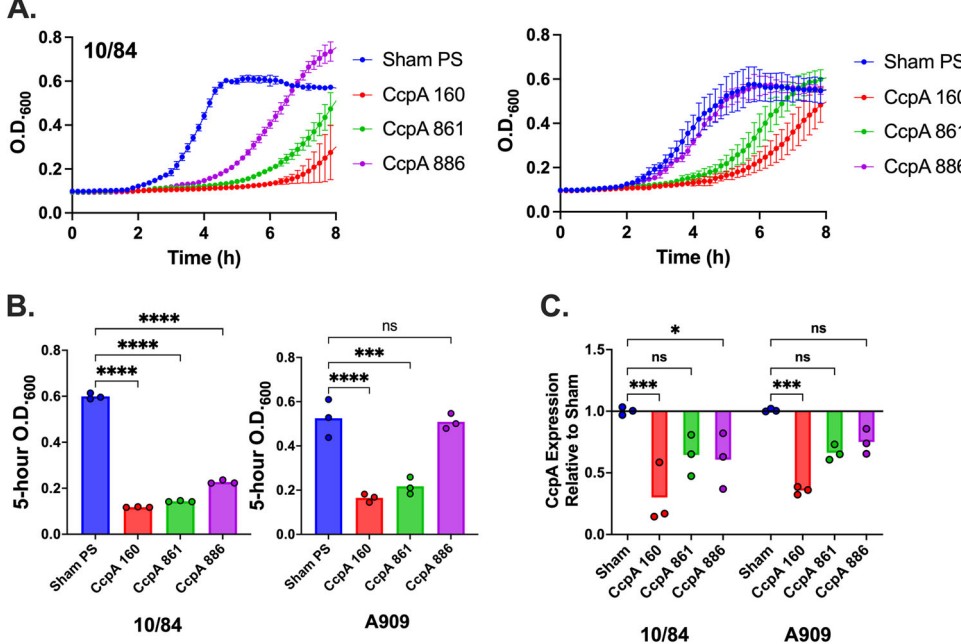

**Fig. 4 CRISPRi allows targeted knockdown of essential GBS genes.** CRISPRi against the *ccpA* gene, which has a significant contribution to GBS fitness and cannot be deleted in A909 grown in tryptic soy broth, resulted in growth defects that were proportional to the location of the gene targeting site within the coding sequence (**A**, $n = 3$ biological replicates, each performed in technical triplicates; data points and error bars are means across biological triplicates and standard deviations at each time point). O.D.$_{600}$ readings of sham-targeted control and targeted 10/84 and A909 *dcas9* strains with *ccpA* expression defects at 5 h of growth (**B**, ***$p < 0.005$ ****$p < 0.001$, one-way ANOVA with Bonferonni correction; $n = 3$ biological replicates, each performed with three technical replicates). qRT-PCR of cDNA using a 5' *ccpA* coding sequence primer pair (normalized to expression of the housekeeping gene *recA*) showed target gene expression knockdown (**C**, ****$p < 0.001$, one-way ANOVA with Bonferonni correction; $n = 3$ biological replicates, each performed with three technical replicates). ns = not significant.

mutations approximately 2500-bp apart), we did not create a *dcas9* revertant.

All our mutant and revertant strains were confirmed through whole-genome sequencing to bear the intended mutations. Whole genome sequences showed that none of the new variants had unintended mutations expected to alter function. The A909 *scas9* and Δ*cas9* variants and revertants did not redemonstrate the prophage excision found in A909 *dcas9*. In tryptic soy broth, the new variants showed sigmoidal growth kinetics indistinguishable from WT and revertant controls (Fig. 6C).

Introduction of p3015b with a protospacer that targets a genomic site next to a NGG PAM should result in lethal chromosomal cleavage in WT and revertant GBS but should be tolerated in *dcas9*, *scas9*, and Δ*cas9*. Additionally, transformation with p3015b with a nontargeting sham protospacer—without a complementary sequence in the GBS genome—should result in successful transformation of all GBS strains. Finally, *scas9* strains should not show changes in βHC production and hemolysis when transformed with p3015b bearing targeting protospacers against the *cyl* operon or *covR*.

In Fig. 6D, conversion of WT *cas9* to *dcas9* or *scas9* restores transformation efficiency, but all revertant strains behave identically to WT, with no colonies appearing after attempted transformation with a *cyl* targeting p3015b plasmid. In vitro hemolysis assays using the new variants, with both chromosomal gene targets repressed, demonstrate expected phenotypes, with *cyl* repression decreasing CNCTC 10/84 hemolysis while targeting of *covR* in A909 increases hemolysis in *dcas9* but not *scas9* strains (Fig. 6E).

**RNA-seq of Cas9 variants shows PAM scanning to influence genome-wide transcription.** If the PAM scanning function of Cas9, which is dependent on amino acid residues Arg1339 and Arg1341, were responsible for transcriptomic influences, both the

*scas9* variant and the Δ*cas9* deletion strain should show similar effects on genome-wide transcription. Furthermore, the *dcas9* variant and the WT strain—both with functional PAM scanning domains—should have similar transcriptomes.

Because of the prophage deletion in A909 *dcas9*, which could lead to confounding genome-wide transcriptional perturbations in this strain, we elected to focus on CNCTC 10/84 for these analyses. We grew CNCTC 10/84 WT, *dcas9*, *scas9*, and Δ*cas9*, without p3015b or other plasmid, in nonselective tryptic soy broth. Triplicate biological replicates were used for total RNA purification at O.D.$_{600}$ 0.6 (mid-log growth) and O.D.$_{600}$ 1.2 (early stationary growth).

RNA-seq data were normalized and evaluated for relationships between the PAM scanning *cas9* variants (WT and *dcas9*) and the non-scanning variants (*scas9* and Δ*cas9*). Figure 7A shows upset plots (histogram-based analogs of Venn diagrams) that quantify genes whose transcription differed significantly, defined as log$_2$ fold-change $\geq 1$ or $\leq -1$ and adjusted $p \leq 0.05$ using the Benjamini–Hochberg algorithm in DESeq2[72,73], in pre-selected comparisons between variants at both mid-log and early stationary growth. See Supplementary Data 2 for DESeq2 output data.

Across both phases of growth, more genes showed differential transcription in comparisons between PAM scanners and non-scanners, and the PAM scanner v. non-scanner comparisons also showed the greatest extent of overlap in the genes affected (see red bars in Fig. 7A). The high correlation between genome-wide differential transcription is illustrated in the heatmap in Fig. 7B, where the mean Pearson correlation $R^2$ was 0.45 (standard deviation 0.17) when read-normalized transcriptome alignments were compared between two scanner v. non-scanner conditions but decreased to 0.09 (standard deviation 0.10; $p < 0.0001$ by two-tailed $t$ test) when there was discordance between the scanner and non-scanner comparisons (for example if a scanner/non-scanner pair was correlated with a scanner/scanner pair). See

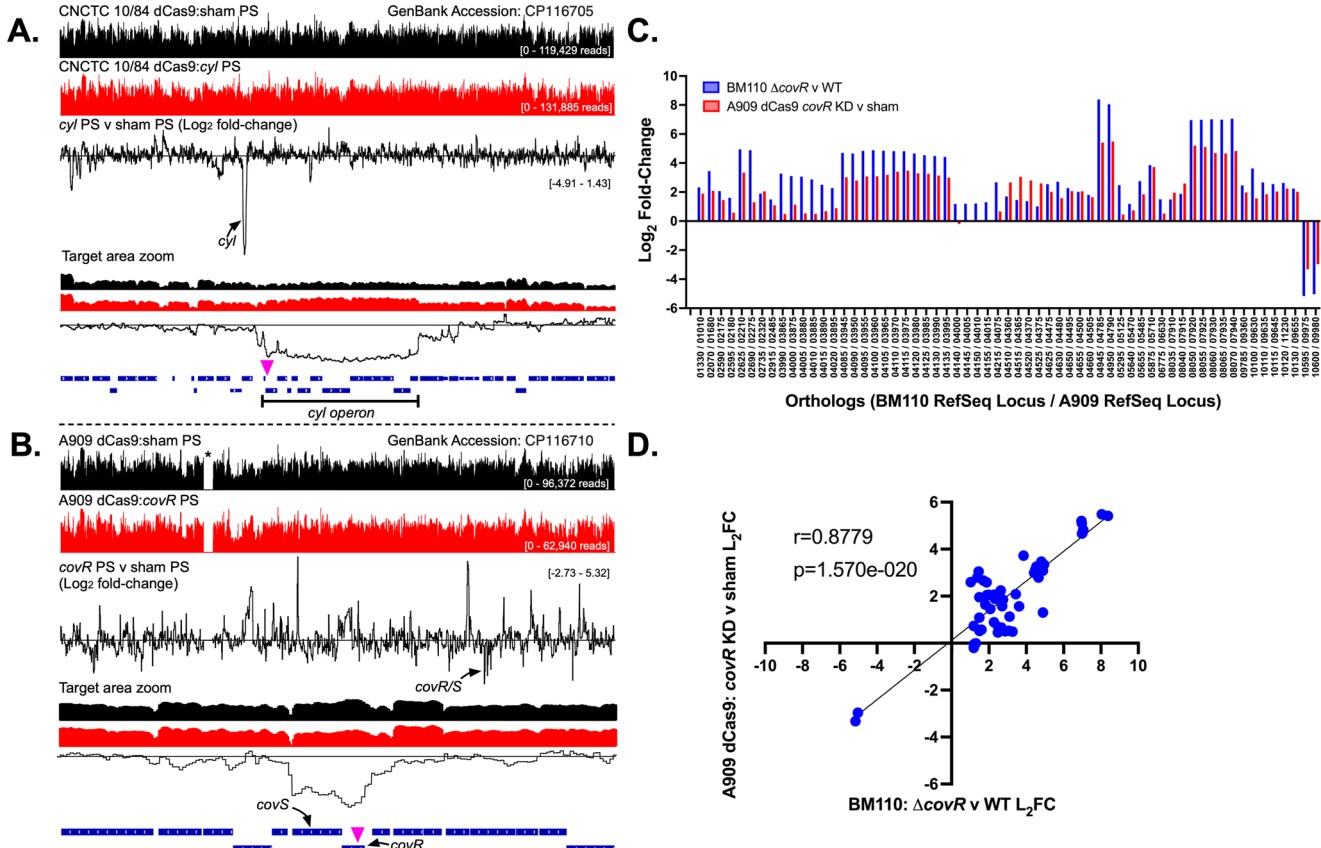

**Fig. 5 RNA-seq of chromosomal CRISPRi targeting shows on-target downregulation and expected downstream effects of transcription factor repression.** Normalized, genome-wide RNA-seq alignment data are shown for 10/84 *dcas9* with *cyl* CRISPRi (**A**) and A909 *dcas9* with *covR* CRISPRi (**B**). Non-targeted sham p3015b control strains (black histograms) were used for comparison to targeted strains (red histograms) in both cases. Triplicate biologic replicates, grown to O.D.$_{600}$ = 1.2, were RNA sequenced from all four strains; alignment data is from all three replicates combined. Log$_2$ fold-change data are shown in which the experimental and control alignments were compared. The zoomed insets display the region of CRISPRi targeting, with the site of the protospacer complement indicated by violet triangles. For A909 *dcas9* with *covR* CRISPRi, orthologous gene expression from RNA-seq was compared to published data defining the *covR* operon in GBS strain BM110[71] (**C**, **D**).

Supplementary Fig. 6 for a matrix of the individual correlation scatter plots. Principal component analysis of genome-wide transcript profiles showed clustering of the scanner/non-scanner comparisons at both growth phases (Fig. 7C).

Genes affected by elimination of Cas9 PAM scanning were distributed throughout the genome without clear regional interrelationships (see Discussion). We used gene set enrichment analysis (GSEA) to evaluate whether genes most affected by Cas9 PAM scanning share functional roles[74,75]. CNCTC 10/84 cluster of orthologous gene (COG) categories were assigned using eggNOG-mapper[76,77], and these gene groupings were used in combination with read depth-normalized, equivalently scaled RNA-seq data to determine COGs whose expression was significantly increased or decreased in the PAM scanner v. non-scanner comparisons.

Figure 7D shows GSEA results. The software reports normalized enrichment scores for each analysis, which are cross-comparable quantifications of significant over- or under-expression of genes in a functional category compared to the rest of the dataset, normalized for the size of each category[74] (see Supplementary Fig. 7). Figure 7D shows normalized enrichment scores (with bars terminating at mean values) for all scanner v. non-scanner comparisons at both growth timepoints. We also present mean false-discovery rate (FDR) q values for each of the four comparisons at each timepoint. Using a FDR q value cutoff of 0.10 revealed two COG categories at O.D. 0.6 and three at O.D. 1.2 for which all scanner v. non-scanner comparisons showed either

relative enrichment or depletion. G (nucleotide metabolism/transport) and V (defense) genes showed significantly decreased expression in the PAM non-scanner variants at both growth phases. F (carbohydrate metabolism/transport) showed enrichment in the non-scanner variants at both growth phases but was only statistically significant at O.D. 1.2.

Finally, in order to hone the subset of genes within each category driving the significant expression differences between PAM scanners and non-scanners, we used the leading edge analysis function within GSEA to identify core genes within each of these three COG categories whose over- or under-expression drove divergence between the variants[74]. These genes, whose normalized expression is illustrated in the heatmaps of Fig. 7E and whose gene loci tags can be found in Supplementary Data 3–6, can be considered the subset of CNCTC 10/84 genes whose expression is consistently altered when Cas9 PAM scanning is eliminated. We performed confirmatory RT-qPCR on cDNA from RNA extracted from GBS Cas9 variants. These results are presented in Supplementary Fig. 8, which confirms expected findings from RNA-seq (see Supplementary Table 3 for primer sequences).

## CNCTC 10/84 cas9 variants show differences in resistance to oxidative stress challenge but equivalent virulence in a murine intraperitoneal injection/sepsis model. Previous publications describe significant differences in GBS virulence in animal models

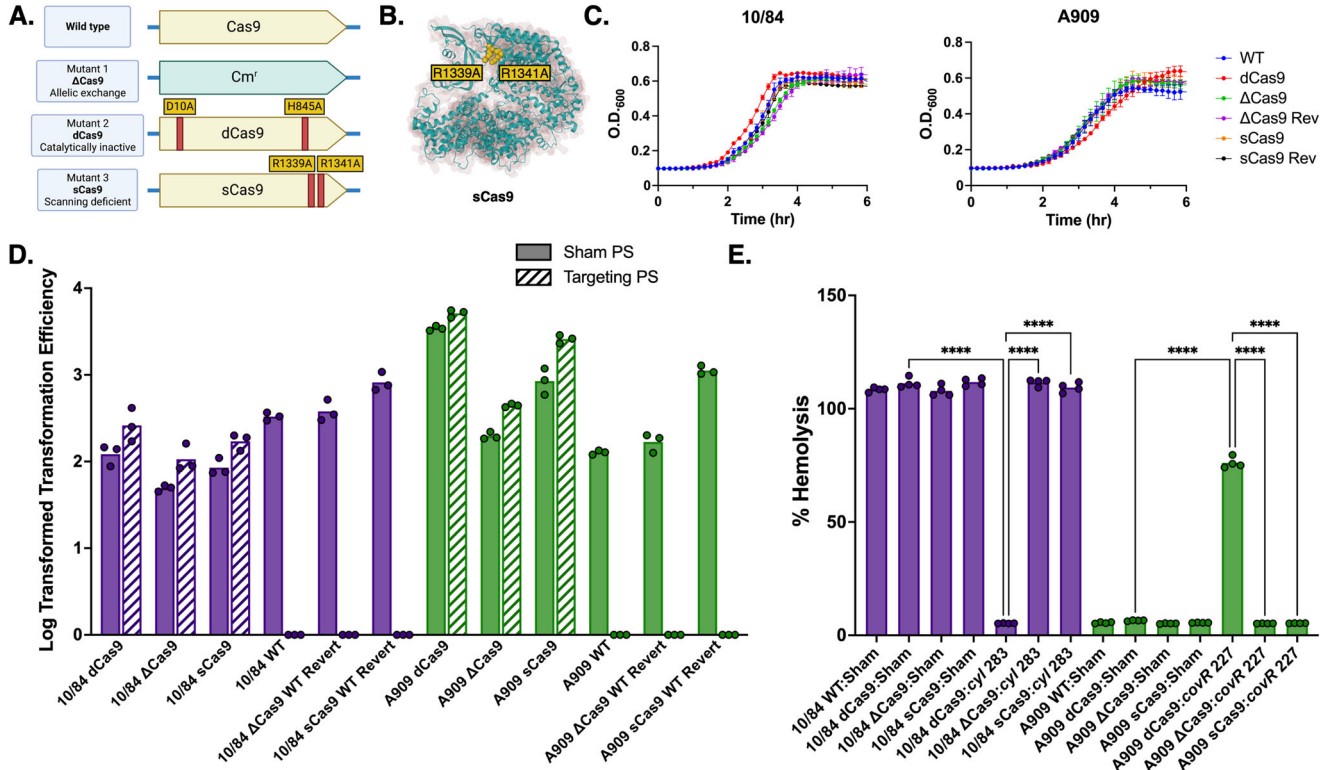

**Fig. 6 GBS Cas9 targeted mutagenesis design and phenotypic features.** We used allelic exchange mutagenesis approaches to generate a Δ*cas9* gene deletion mutant and PAM scanning deficient *scas9* with R1339A and R1341A in CNCTC 10/84 and A909 (**A**, **B**). The Cas9 variants showed equivalent growth kinetics in tryptic soy broth (**C**, $n = 3$ biological replicates, each performed in technical triplicates; data points and error bars are means across biological triplicates and standard deviations at each time point). p3015b with a targeting protospacer (PS) directed against the *cyl* operon is incompatible with WT GBS, in which it causes lethal double stranded DNA cleavage, but it can be introduced into *dcas9*, *scas9*, and Δ*cas9*. p3015b with a nontargeting sham protospacer can be successfully introduced into all Cas9 variants (**D**). Each transformation was performed in triplicate biological replicates, from which all data are shown. Confirmatory hemolysis assay results in both strains matched phenotype expectations from the regulatory model (**E**, $n = 4$ biological replicates from which all data are shown, ****$p < 0.001$, one-way ANOVA with Bonferroni correction).

when Cas9 function is altered by deletion of the *cas9* gene or the CRISPR array[31,32,78]. Given the transcriptomic differences between PAM scanning and non-scanning Cas9 variants in CNCTC 10/84, we wanted to know whether these variants would show phenotypic differences under growth conditions that applied environmental challenges relevant to infection. First, we tested growth of our variants under oxidative stress from 5 mM porcine hemin added to growth media in vitro. We also applied our variants to a mouse intraperitoneal injection model that results in bacteremia and multi-organ system dissemination[79].

In the oxidative stress challenge in 5 mM hemin, we observed small but reproducible changes in maximum O.D.$_{600}$, with WT and *dcas9* growing to higher density than *scas9* and Δ*cas9* comparators (Supplementary Fig. 9).

Following infection with $2 \times 10^8$ colony forming units (CFU) of washed, stationary phase WT, mutant, or revertant CNCTC 10/84, adult BALB/cJ mice ($n = 7–8$) were serially observed over 48 h for death or pre-defined moribund signs. Pilot experiments with lower bacterial loads resulted in rapid clearance with sterile organs at harvest, while higher loads caused complete mortality (regardless of the strain used) within 24 h, concerning us that we might miss meaningful differences in virulence. $2 \times 10^8$ CFU caused consistent dissemination and signs of illness while still permitting 48 h of observation.

In our experiment, none of the infected mice died spontaneously. Those shown as nonviable in Fig. 8A all met the sole humane endpoint criterion of weight loss more than 20% of baseline. Summary weight statistics for all cohorts are shown in

Fig. 8B. Comparisons were nonsignificant between *dcas9* and WT, Δ*cas9* and its revertant control (Δ*cas9* Rev), and *scas9* and the *scas9* revertant (*scas9* Rev). Regardless of clinical appearance, all cohorts were dissected at 48 h post-infection. Sterilely collected blood, lung, spleen, and PBS peritoneal lavage samples were serially diluted and plated on GBS-selective plates for next-day CFU quantification. Again, we did not observe significant differences in bacteremia, organ seeding, or peritoneal persistence in the same comparisons listed above (Fig. 8C).

## Discussion
Molecular technologies based on CRISPR-Cas systems have had a transformational effect on biology research over the past decade[80]. At the center of the CRISPR-Cas revolution is the ability to flexibly direct a modular, modifiable protein to a specific nucleotide sequence. This fundamental advance has driven tremendous innovation and a wide array of new applications and discoveries.

In GBS research, the native CRISPR-Cas9 system has become a focus of study and the basis of new tools for pathogenesis research, such as using the highly variable CRISPR array as a strain-level barcode for lineage tracking in models of disease[29,30].

The genome-wide transcriptional influence exerted by the GBS CRISPR-Cas9 system is unusual and invites investigation. Off-target effects by Cas9 have been carefully and thoroughly examined in eukaryotic cell systems, where they can be problematic in the context of genome editing, CRISPR library preparation, and other applications[81]. Nonspecific transcriptional effects have also

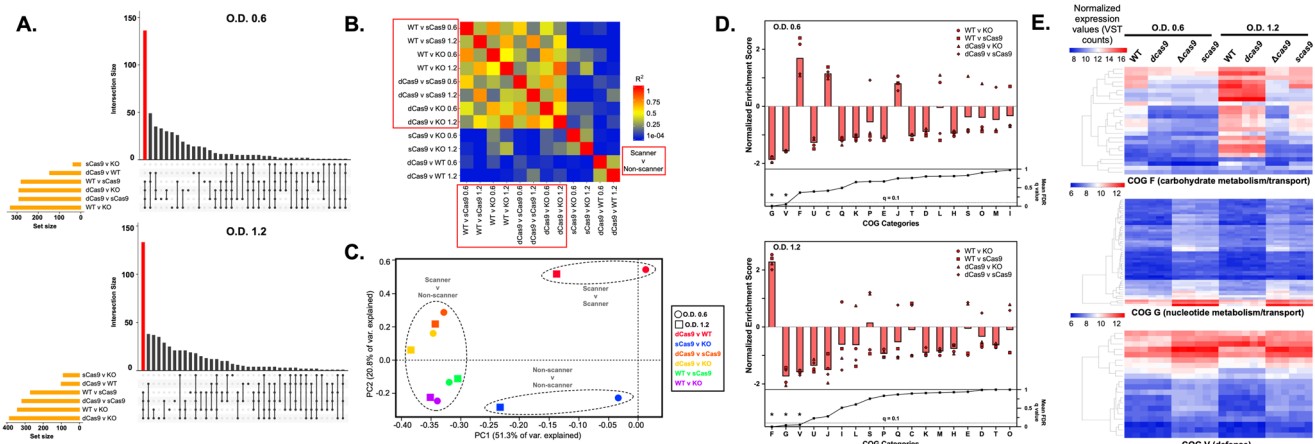

**Fig. 7 PAM scanning influences genome-wide transcriptional modulation in GBS.** RNA-seq was performed on CNCTC 10/84 Cas9 variants in triplicate biological replicates at two phases of growth (mid-log, O.D.$_{600}$ = 0.6; early stationary, O.D.$_{600}$ = 1.2). Upset plots of DESeq2 analysis after sequence data normalization and strain alignment showed that the majority of differentially regulated genes emerged from comparisons between PAM non-scanning strains ($\Delta cas9$ or $scas9$) and PAM scanning strains (WT or $dcas9$), which are highlighted in the red bars (**A**). Pearson correlation between scanner v. non-scanner comparisons was significantly higher than comparisons between scanner v. non-scanner and either scanner v. scanner or non-scanner v. non-scanner (**B**, two-tailed Student's $t$ test on $R^2$ values < 0.0001). Clustering of scanner v. non-scanner comparisons, at both growth time points, is illustrated in the principal component analysis of whole-genome RNA-seq data in (**C**). Gene set enrichment analysis of scanner v. non-scanner comparisons at both time points demonstrated decreased expression of genes in cluster of ortholog (COG) functional groups G (nucleotide metabolism/transport) and V (defense) in PAM scanning strains relative to non-scanners. COG group F (carbohydrate metabolism/transport) showed more expression in the scanners than non-scanners at both time points but was only significantly different (with a false discovery rate ≤ 0.10) at early stationary phase (**D**). Leading edge analyses of COG groups F, G, and V are illustrated as heat maps (**E**), where the values plotted are variance stabilizing transformation (VST) counts, which give a relative measure of gene expression levels that are normalized (for sequencing depth) and stabilized (for variance across sample replicates) allowing comparisons between different sample conditions[72]. There is no across-row or across-column scaling of these values, such that cross-comparison between rows, columns, and plots are valid.

been described in studies of bacterial CRISPR-Cas9 systems. However, these transcriptional influences have generally been traced back to partial complementarity between the gRNA and the unintended target[82], or interactions between the Cas9 enzyme and non-canonical small RNA molecules, which were shown in *Francisella novicida* to lead to predictable off-target transcriptional modulation[83]. In GBS, however, the extent and diversity of genes influenced by Cas9 suggest an alternative, less specific mechanism.

We considered these observations in light of what is known about Cas9 interactions with the chromosome, independent of the sequence of a bound gRNA molecule. We thought it possible that promiscuous scanning by Cas9 as it binds and unbinds from NGG PAM sites might contribute to changes in DNA accessibility to transcription machinery and other regulators, which might result in up- or downregulation of distinct regions or gene subsets. Such a mechanism is compatible with the observation by Spencer et al. that deletion of the *cas9* gene in GBS strain COH1 has a significant effect on genes regulated by a specific transcription factor, CiaR[31]. As a test of this hypothesis, we examined genome-wide transcriptomic data in GBS mutants that either lacked *cas9* altogether or had targeted single nucleotide polymorphisms rendering the Cas9 enzyme unable to bind PAM sites (*scas9*) or unable to cleave DNA after gRNA complementary sequence recognition (*dcas9*).

Correspondence between the transcriptional effects observed in these Cas9 variants suggests that PAM scanning is a significant contributor to CRISPR-Cas9 genome-wide influences on GBS transcription. The genes that were differentially transcribed between PAM scanning and non-scanning variants are distributed across the genome. We considered multiple possible variables that might lead to genes being more or less transcribed when Cas9 is either capable or incapable of performing its PAM-scanning function. We assessed for relative transcriptomic

covariance with GC-content, atypical nucleotide content, density of NGG PAM sequences within or upstream of gene coding sequences, and a systematic search for partial matches to protospacers within the native CNCTC 10/84 CRISPR array.

None of these analyses yielded a positive signal (negative results available upon request). Instead, we found clustering of the most affected genes in functional categories defined by involvement in carbohydrate and nucleotide metabolism and bacterial cellular defense. Since many of these critical cellular processes are co-regulated by transcription factors that sense and respond to environmental flux, we continue to suspect that Cas9 steric hinderance of transcriptional machinery during its PAM scanning process is the most likely explanation for the transcriptional effects we report here. The influence of PAM scanning on Cas9 bacterial genome-wide transcriptional effects has not, to our knowledge, been previously reported. Future work may be directed at determining whether these same effects occur in bacteria other than GBS. It is also important to acknowledge that Cas9-PAM interactions are complex and likely consist of three-dimensional diffusion (DNA attachment, detachment, distal reattachment) and one-dimensional diffusion (lateral movement along DNA from one PAM to another nearby PAM)[84]. This complex, multidimensional diffusion and interaction model is consistent with variable, genome-wide transcriptional effects, possibly with multiple, competing, and variable influences.

The other contribution of this study is proof-of-concept and initial characterization of GBS dCas9-based CRISPRi. Since the original demonstration that catalytically inactive Cas9 could be used to target transcription of specific genes[37], CRISPRi has emerged as a powerful discovery tool in a variety of experimental systems, including in studies of bacterial cell biology and virulence[26,27,38,39,82,85–90].

The CRISPRi system we present in GBS is simple to use. In a strain that has been modified to constitutively express dCas9,

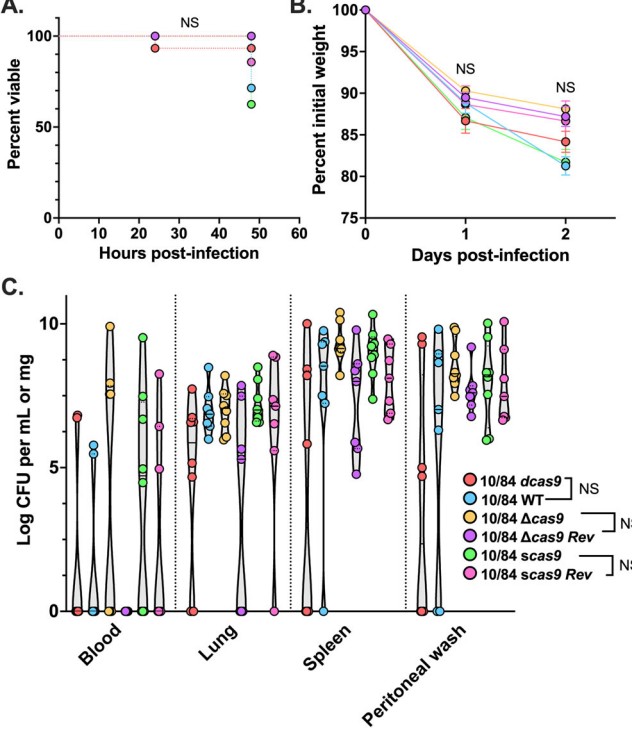

**Fig. 8 Virulence phenotypes are equivalent in an adult mouse sepsis model with CNCTC 10/84 Cas9 variants.** Adult BALB/cJ female mice ($n = 7$–8) who received intraperitoneal injection of $2 \times 10^8$ CFU of GBS were monitored for death or demonstration of predefined humane endpoints. None of the infected mice died spontaneously, but multiple met the endpoint of $\geq$20% weight loss from baseline and are shown as nonviable prior to planned dissection at 48 h (**A**, Mantel-Cox test; **B**, one-way ANOVA with Bonferroni correction for multiple pre-planned comparisons diagrammed in the legend). Quantification of GBS CFU density in blood, organs, and peritoneal washes collected immediately following euthanasia revealed no significant differences in intraperitoneal persistence or systemic dissemination between pre-planned comparisons (**C**, one-way ANOVA with Bonferroni correction). NS = not significant.

introduction of an easily purified, broad host-range expression plasmid with a 20-bp targeting cassette cloned into a pair of BsaI restriction sites leads to reliable downregulation of expression from essential or nonessential genes on the chromosome or another plasmid. As in other CRISPRi systems, maximal effect is achieved when the target sequence complementary to the sgRNA is on the antisense strand of the gene and the target is situated in the gene promoter or near the 5' end of the coding sequence. Because our approach uses precisely altered, native GBS dCas9—expressed from the chromosome—we retain the genome-wide PAM scanning effects that exist in WT.

Previous publications have described phenotypic and virulence changes in GBS when *cas9*[31] or the CRISPR array[32,33] are altered. In CNCTC 10/84 Cas9 variants, we observed subtle phenotypic changes in an oxidative stress challenge in vitro (Supplementary Fig. 9) that corresponded to transcriptomic patterns, with shared growth phenotypes between scanning and non-scanning mutants. However, we observed no change in major outcomes in a murine sepsis model (Fig. 8). There were differences between our experiments and those already published. Spencer et al. (2019) used mouse models of bacteremia and meningitis and a vaginal colonization competition assay, performed with a COH1 WT strain and Δ*cas9* mutant[31] and reported differences in density of GBS brain invasion and competition indices in the vaginal tract

with no differences in bloodstream or lung bacterial burdens. Another group used a hypervirulent piscine GBS variant, which—compared to our non-piscine strains—was extremely aggressive in mouse models of bacteremia and subsequent meningitis, virulence traits that were attenuated in a ΔCRISPR mutant in the same background[32,33]. Surveying the data across all these studies, we find it probable that the transcriptomic changes in GBS with altered Cas9 scanning capacities lead to changes in sensitivity to immune host defenses during colonization and infection, but that these changes are subtle and may not always correspond to severe outcomes such as death.

Our experiments with CRISPRi targeting essential genes (Fig. 4) show that growth delays, rather than complete growth suppression, typically result. The extent of growth delay seems to be a function of target position along the gene, with targets nearer to the start of the coding sequence resulting in more severe growth suppression. This observation argues against stochastic mutations in the plasmid protospacer, *dcas9*, or the gene target itself as an "escape" mechanism, as these events would not be expected to relate to the gRNA target site. We speculate that the delayed growth we observe for knockdowns of these essential genes may reflect gradual accumulation of enough full-length transcript to exceed a threshold needed for growth, and that gRNA targets near the start of a gene may impede this accumulation more than targets nearer the terminus.

We also saw that targeting essential genes very near their termini had variable effects between the two strains we studied (see the differences between the sgRNA targeting position 886 in Fig. 4). The *ccpA* coding sequence is 1005 bp long, so the gRNA targeting position 886 is in the terminal 15% of the gene. As suggested above, we believe that in edge cases where RNA polymerase collides with dCas9 at the very end of a gene, a modest amount of full-length transcript may end up being transcribed (an analogy might be a cryptic promoter that inefficiently generates some transcript). CNCTC 10/84 and A909 show very different growth phenotypes across a range of cell stress conditions. We hypothesize that CcpA requirements may differ between the two strains and that our expression titration by moving the gRNA target identifies experimentally different *ccpA* expression thresholds for the two strains at which knockdown leads to major growth delay.

In addition to straightforward applications to screen candidate genes for effects on growth or other cellular characteristics, we envision additional strategies for using CRISPRi and other dCas9-based experiments to answer challenging questions about GBS biology. CRISPR-Cas activator (CRISPRa) systems that upregulate targeted genes, CRISPRi library-based approaches to evaluating gene fitness among multiple candidate genes, inducible CRISPRi systems, and dCas9-based nuclear colocalization imaging and coimmunoprecipitation are likely feasible using the targeted mutagenesis and sgRNA expression systems described here.

We have presented a plasmid-based sgRNA expression system, which would require administration of antibiotics to ensure retention of the plasmid by GBS, which would otherwise cure the exogenous DNA and revert to a *dCas9* phenotype likely indistinguishable from WT. However, chromosomal integration of the sgRNA expression cassette would be straightforward with existing techniques[91] and we would expect this strategy to allow CRISPRi-based interrogation of specific genes in longitudinal in vivo studies, such as investigations of persistent GBS colonization.

Together, the results presented here add to a growing understanding of GBS CRISPR-Cas biology, the role that Cas9 plays in modulating the genome-wide transcription profile, and how the precise, flexible power of Cas9 targeting can be used in GBS to examine the relationships between gene expression and phenotype in a variety of experimental settings.

## Methods

**Ethics statement**. Animal experiments were performed under approved IACUC protocols at the University of Pittsburgh (20016575). Adult phlebotomy for hemolysis assays was conducted under a University of Pittsburgh approved IRB protocol (19110106). Human participants agreed to voluntary blood draws without collection of identifiable information, following review of risks, benefits, and alternatives, including nonparticipation.

**Statistics and reproducibility**. Standard statistical tests on data other than genome-level sequencing results were performed using GraphPad Prism (MacOS version 9). Only publicly available commercial and non-commercial software was used for data analysis. There was no development of custom software or algorithms. Next-generation sequencing alignments were performed with Bowtie2 (version 2.4.5). Raw read alignments were quantified with HTSeq (version 2.0.2). DESeq2 (version 3.16) was used for statistical analyses of RNA-seq data. The following DeepTools (version 3.5.1) analysis algorithms were performed on a dedicated instantiation of the Galaxy bioinformatics system maintained and run by the University of Pittsburgh Center for Research Computing: multiBAMSummary, computeMatrix, plotCorrelation, and plotPCA. Upset plots were generated using the UpsetR package (version 1.4.0). GBS gene COG categories were determined with eggnog-Mapper (version 2.1.9). Gene set enrichment analyses were performed with the GSEA software package (version 4.3.2). Specific genome-level bioinformatic statistical testing is described in the relevant sections below.

GBS experiment sample sizes were determined based on experience with the assays, which showed significant differences when duplicate or triplicate technical replicates were used across two biological replicates. Mouse experiment sample sizes were based on power calculations to detect 80% differences in bacterial load with a power of 80% and alpha = 0.05. All numeric data for figures throughout this article are available in Supplementary Data 7 (except for RNA-seq data, which are publicly available in the Gene Expression Omnibus database).

**Bacterial strains and growth conditions**. GBS strains A909 (serotype Ia, sequence type 7) and CNCTC 10/84 (serotype V, sequence type 26), both of which were originally isolated from infected neonates[41,53,92], and their derivatives were grown at 37 °C (or 28 °C when the temperature-sensitive pMBsacB plasmid was present and extrachromosomal) under stationary conditions in tryptic soy (TS) medium (Fisher Scientific cat. # DF0370-17). GBS was supplemented with 5 μg/mL erythromycin or 1 mg/mL kanamycin as needed for selection. *E. coli* strain DH5α was grown at 37 °C (or 28 °C with extrachromosomal pMBsacB present) with shaking in Luria-Bertani (LB) medium (Fisher Scientific cat. # DF9446-07-5) supplemented with 300 μg/mL erythromycin or 50 mg/mL kanamycin as needed for selection.

**Generation of GBS cas9 variant mutants dcas9, scas9, and Δcas9, and revertant controls**. GBS mutants in A909 and CNCTC 10/84 backgrounds were generated with the temperature-sensitive, sucrose-counterselectable mutagenesis plasmid pMBsacB using procedures previously described in detail[91]. Mutagenesis cassettes for cloning into pMBsacB linearized through double restriction enzyme digestion with NotI and XhoI were generated by PCR from appropriate templates, followed by Gibson assembly to clone the assembled cassettes into the plasmid.

*dcas9* strains were generated through two mutagenesis cycles to produce the separate D10A and H840A required. Custom synthesized DNA (GenScript, Piscataway, NJ) bearing the N- and C-terminal mutations were amplified with dcas9_mut1_ F/R and dcas9_mut2_F/R (Supplementary Table 1).

*scas9* strains were generated by amplifying appropriate GBS genomic DNA with primers scas9_US_F/R and scas9_DS_F/R. The resulting two fragments had the desired R1339A/R1341A mutations incorporated into a central homology region such that Gibson assembly led to generation of a single fragment with the correct sequence.

*Δcas9* was similarly generated by amplifying upstream and downstream GBS homology arms, from the appropriate strain's genomic DNA, using xcas9_US_F/R and xcas9_DS_F/R. The chloramphenicol resistance gene, *cat*, was amplified from the plasmid pDC123[93] using primers xcas9_cat_F/R.

Cloning steps were followed by transformation into chemically competent *E. coli* DH5α (NEB cat. # C2987H) and correct mutagenesis construct sequences were confirmed by Sanger sequencing. Next, the plasmids were purified by miniprep and transformed into electrocompetent GBS as previously described[91,94,95].

Transformed GBS were transitioned to 37 °C, the pMBsacB non-permissive temperature, under erythromycin selection. Correct "single-cross" insertion of pMBsacB into the genome was confirmed by PCR and Sanger sequencing. Next, sucrose counterselection was applied to select for "double-cross" mutants and revertants. Revertant strains were saved for *scas9* and *Δcas9*. Because *dcas9* strains were constructed in two temporally separate steps, we did not generate *dcas9* revertants.

To ensure that no unintended off-site mutations were affecting bacterial phenotypes, mutant and revertant strains were used to generate sequencing libraries with IDT 10 bp UDI indices then sequenced to 200 Mbp using 2 × 151 bp paired end reads on an Illumina NextSeq 2000 platform. Demultiplexing, quality control and adapter trimming was performed with bcl-convert (v3.9.3) followed by variant calling and analysis.

**Protospacer cloning into p3015b**. p3015b is a derivative of vpL3004, which is an erythromycin-resistant, low copy number, broad host range cloning plasmid originally used for CRISPR-Cas aided recombineering in *Lactococcus lactis*[17]. We used Gibson assembly to insert a synthesized, double-stranded GBS-optimized sgRNA cassette (Genscript custom order) with an upstream *xyl/tet* promoter that constitutively expresses the sgRNA. The cassette harbors two internal BsaI restriction enzyme recognition sites that permit rapid, in frame protospacer cloning (Supplementary Fig. 2) through a process adapted from concepts outlined by Jiang et al.[25]. The cassette is flanked by two rho-independent transcription terminators.

After p3015b linearization with BsaI, the plasmid backbone was purified by agarose gel extraction. Each protospacer was generated as two single-stranded, complementary oligonucleotides (Integrated DNA Technologies, custom orders) matching 20-nt of the target site and designed to generate compatible BsaI single-stranded overhangs once annealed (Supplementary Table 2).

10 μM solutions of each protospacer oligonucleotide were end-phosphorylated with T4 polynucleotide kinase (NEB cat. # M0201, Ipswich, MA) for 30 min in the presence of 1x T4 ligase buffer then annealed by heating to 95 °C in a heat block which was subsequently allowed to cool to room temperature. Next, 1:10 dilutions of the phosphorylated, annealed protospacers were cloned into the BsaI-digested p3015b with T4 DNA ligase (NEB cat. # M0202) at 16 °C overnight, then transformed into competent DH5α *E. coli*. Proper protospacer cloning was confirmed by Sanger sequencing. Miniprepped samples of purified p3015b:protospacer were then used for transformation into electrocompetent GBS strains[91,94,95].

**Determination of transformation efficiency**. Electrocompetent GBS was prepared following methods outlined in Framson et al.[95]. Briefly, single colonies were grown in 30 mL of M17 broth with 0.5% glucose overnight before being used to seed 150 mL of prewarmed M17 broth with 0.5% glucose and 0.6% glycine for strain 10/84 and 2.5% glycine for strain A909. Cultures were grown to mid-log phase, determined by an absorbance of 0.6–0.8 at 600 nm, before pelleting at $4000 \times g$ and two washes in ice cold 10% PEG6000. After the final wash, samples were resuspended thoroughly in 1 mL of ice cold 25% PEG600 + 10% glycerol. 50 μL aliquots were stored at −80 °C.

The concentration of viable CFUs in each batch of electrocompetent GBS was measured. Then, 50 μL volumes of competent cells were transformed in triplicate with 5 μL of 150 ng/μL p3015b harboring either a sham or chromosomal targeting protospacer. Transformation plates were allowed 48 h to grow before colony counting. Transformation efficiency was calculated as a ratio of number of transformed colonies by total number of viable CFUs in the electrocompetent cell preparation.

**Bacterial growth curves**. Single colonies from streaked plates were suspended in 50 μL of PBS, pipetted until homogenous, and normalized to an absorbance of 0.5 at 600 nm. 195 μL of media was then inoculated with 5 μL of PBS suspended colony in a sterile flat bottom 96 well plate. Growth of samples was monitored by absorbance at 600 nm, at intervals of 10 min, during stationary incubation at 37 °C in a BioTek Epoch2 plate reader.

**Luminescence assay for reporter strains**. GBS single colonies were used to inoculate 10 mL cultures of tryptic soy medium containing 1 mg/mL kanamycin and 5 ug/mL erythromycin (for retention of plasmids pFfluc and p3015b respectively) and allowed to grow overnight at 37 °C. Cultures were allowed to cool to room temperature before vortexing and normalization to O.D.$_{600}$ = 1.

Black-walled, clear-bottomed 96 well plates were loaded with 5 μL of culture and 195 μL of tryptic soy media containing 1 mg/mL Kan, 5ug/mL Erm, and 110 ug/mL D-Luciferin. Luminescence and absorbance at 600 nm of each well were measured in sequence, at intervals of 15 min, using a Synergy H1 Plate reader during an 8-hour stationary incubation at 37 °C.

**pFfluc PAM mutagenesis**. Complementary site-directed mutagenesis primers were designed to alter the N of the antisense NGG PAM site at residue 104 of the firefly luciferase gene. PAM site mutagenesis targeted the wobble base of proline to ensure silent mutations. Mutagenesis PCR was performed using NEB Q5 High Fidelity Polymerase (cat. #M0492L). PCR products were digested with 1 μL of DpnI for 1 h at 37 °C. Samples were then GibsonI assembled using NEB HiFi DNA Assembly Master Mix (cat. # E2621L) before transformation into NEB DH5α Chemically Competent *E. coli* (cat. # C2987H). All complete plasmids were confirmed by PCR and Sanger sequencing.

**RNA purification, robotic for RNA-seq**. Three single GBS colonies off petri plates from frozen stocks were seeded into tryptic soy broth and allowed to grow overnight. These cultures were then seeded the following morning into fresh broth and monitored until the O.D.$_{600}$ reached 0.6. At that time, 5 mL was rapidly pelleted and mixed with 1 mL of Qiagen RNAprotect Bacteria Reagent, vortexed, and allowed to incubate for 5 min. The mixture was then re-pelleted and the supernatant removed, followed by freezing at −80 °C.

The remainder of the culture was allowed to grow until O.D.$_{600}$ = 1.2, at which point the storage steps above were repeated. At both timepoints, a small sample of

the culture was used for serial dilutions and plating on tryptic soy agar to confirm equivalent growth and to rule out accidental contamination.

After collection and storage of all samples, RNA was extracted on a 4-channel Hamilton Nimbus robotic pipetting station with an integrated ThermoFisher KingFisher Presto 96-head magnetic purification instrument. The extraction was performed with ThermoFisher MagMAX Viral/Pathogen Ultra Nucleic Acid Isolation reagents (cat. #A42356) according to manufacturer instructions. DNase treatment as described above was performed on all samples, which were then tested by NanoDrop and TapeStation analysis prior to sequence library preparation.

**RNA-seq and analysis**. Strand-specific, dual unique indexed libraries for sequencing on all Illumina platforms were made using the NEBNext Ultra™ II Directional RNA Library Prep Kit for Illumina (New England Biolabs). Manufacturer protocol was modified by diluting adapter 1:30 and using 3 μl of this dilution. Size selection of the library was performed with AMPure SPRI-select beads (Beckman Coulter Genomics, Danvers, MA). Glycosylase digestion of adapter and 2nd strand was done in the same reaction as the final amplification to avoid further cleanups. Sample input for this method was ribosomal RNA reduced RNA. Libraries were assessed for concentration and fragment size using the DNA High Sensitivity Assay on the LabChip GX Touch (Perkin Elmer, Waltham, MA). The library concentrations were also assessed by qPCR using the KAPA Library Quantification Kit (Complete, Universal; Kapa Biosystems, Woburn, MA).

The libraries were pooled and sequenced on an Illumina NovaSeq 6000 using 150 bp PE reads (Illumina, San Diego, CA). Following quality control and barcode trimming, alignment to the corresponding genome was performed using Bowtie2 in paired end mode with a maximum insert size of 600 bp[96]. Raw read counts were estimated for all samples using HTSeq in "union" mode[97]. DESeq2 was then used to perform comparisons between read depth-normalized alignments.

Further genome-wide data manipulations and bioinformatic analyses were performed on a dedicated Galaxy instance[98] maintained by the University of Pittsburgh Center for Research Computing. DeepTools[99] functions were performed to merge and compare normalized BAM alignments (with multiBAMSummary) from triplicate biological replicates. The DeepTools computeMatrix, plotCorrelation, and plotPCA functions were used to calculate and visualize cross-alignment comparisons. Upset plots of differentially expressed genes were generated using the UpsetR package[100].

**Transcriptomic gene set enrichment analysis**. Genome-wide COG categories were determined for CNCTC 10/84 using eggnog-Mapper (version 2.1.9)[76,77]. These COG classifications were then reformatted for the Broad Institute's GSEA software package[74]. Normalized enrichment scores were determined based on read depth normalized raw sequence alignment counts for CNCTC 10/84 WT and cas9 variants, along with false discovery rate q values. Leading edge analyses were performed for COG categories F, G, and V, and GBS genes that were present across all PAM scanner v. non-scanner comparisons for either or both growth phases were extracted and used to generate expression heatmaps with www.heatmapper.ca.

**RNA purification, manual for real time qRT-PCR**. GBS single colonies were used to inoculate 10 mL cultures of tryptic soy media and allowed to grow at 37 °C to an absorbance of 0.6 (mid-log phase) at 600 nm. Cultures were pelleted at 4000 × g for 1 min and resuspended in 1 mL of Qiagen RNAprotect Reagent (cat. #76506). The RNAprotect resuspended sample was allowed to incubate at room temperature for 5 min before pelleting and supernatant removal. Bacterial pellets were stored at −80 °C before RNA extraction.

RNA was purified using Invitrogen RiboPure RNA Purification Kit (cat. #AM1924) according to manufacturer instructions. Samples were bead lysed for 1 h instead of the recommended 10 min to ensure proper lysing of the gram-positive cell wall.

Extracted RNA was then DNase treated with 2 μL of DNase and incubated at 37 °C for 1 h. DNase inactivation buffer was added 1 μL per 10 μL of total volume of DNase reaction and allowed to incubate for 5 min, gently mixing intermittently, before removal by pelleting of the colloid enzyme mixture and withdrawal of the supernatant.

**Real time qPCR determination of gene expression**. cDNA was reverse transcribed from purified RNA using the Applied Biosystems High-Capacity Reverse Transcription Kit (cat. #4368814). Each RNA sample had a reverse transcriptase negative condition as a control. qRT-PCR was performed with Bio-Rad SSOadvanced Universal SYBR Green Supermix (cat. #1725270) and target sequence specific primers. The GBS housekeeping gene recA was used as the normalization control. qPCR reactions were performed on a Bio-Rad CFX96 real time PCR thermocycler. Target gene expression was compared to an orthologous control strain harboring p3015b with a sham nontargeting protospacer. Normalized gene expression was calculated using the Bio-Rad Livak Method. Primer sequences used for qRT-PCR are listed in Supplementary Table 3.

**Hemolysis assay**. GBS single colonies were used to inoculate 10 mL cultures of tryptic soy medium and allowed to grow overnight at 37 °C. Cultures were washed once in PBS and then resuspended in 1 mL of PBS. O.D.$_{600}$ of PBS resuspensions

was then normalized to 1. 5 mL of human blood was drawn by sterile phlebotomy, heparinized, and washed three times in Hank's Balanced Salt Solution (HBSS). Washed red blood cells (RBC) were spun at 200 rpm for 8 min to pellet RBC. 50 μL of RBC pellet was collected and diluted into 5 mL of HBSS to create a 1% red blood cell solution (RBCS). 100 μL of 1% RBCS were combined with 100 μL samples of bacterial PBS suspensions in a V-bottom 96 well plate and allowed to incubate for 1 h at 37 °C in a 5% CO2 incubator. The plate was then spun at 2000 rpm at 4 °C for 5 min and supernatants were transferred to a flat bottom 96 well plate. Hemoglobin release was determined by measuring absorbance of the supernatants at 415 nm and percent hemolysis was defined relative to a positive control of RBC treated with Triton-X-100.

**Mice**. BALB/cJ mice were purchased from Jackson Laboratories. All mouse experiments were performed in an American Association for the Accreditation of Laboratory Animal Care-accredited animal facility at the University of Pittsburgh and housed in accordance with the procedures outlined in the Guide for the Care and Use of Laboratory Animals under an animal study protocol approved by the Institutional Animal Care and Use Committee of the University of Pittsburgh. Mice were housed in specific pathogen-free conditions.

**Adult mouse GBS sepsis model**. The adult mouse GBS sepsis model was performed as previously described[79]. Briefly, 8- to 12-week old female BALB/cJ mice were injected intraperitoneally with $2×10^8$ CFU of PBS-washed, stationary phase bacteria suspended in 500 μL of PBS. Each bacterial strain was injected into 7-8 mice. The weights of the mice were measured at the time of injection and monitored daily. Mice found to have lost 20% or more of baseline weight were considered nonviable and were euthanized. At 48-hours post infection, tissues were harvested from the mice, homogenized and serially diluted, followed by plating on GBS-specific CHROMAgar plates and growth overnight to determine bacterial load. We also plated serial dilutions of a 500 μL intraperitoneal PBS lavage instilled then withdrawn at the time of dissection.

**Generation of images**. Some of the images (Figs. 1, 6A) in this article were created using BioRender.com. Others (Supplementary Figs. 1–3) were created with Geneious Prime software (distributed by Biomatters).

**Reporting summary**. Further information on research design is available in the Nature Portfolio Reporting Summary linked to this article.

## Data availability

Illumina whole-genome sequence and RNA-seq data are publicly available on NCBI under umbrella BioProject accession PRJNA949218. Remaining data supporting the findings of this study are available within the paper and in Supplementary Data 7.

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

## Acknowledgements

Financial support was provided by NIH/NIAID R21AI147511 to T.A.H. and H.T.; K08AI132555 to T.A.H.; a pilot award from the University of Pittsburgh Institute for Infection, Inflammation, and Immunity in Children to T.A.H.; a pilot award from the Richard King Mellon Institute for Pediatric Research to T.A.H.; and a trainee grant from the UPMC Children's Hospital of Pittsburgh Research Advisory Council to K.P.G. The funders had no role in study design, data collection and analysis, decision to publish, or preparation of the manuscript. We are grateful to the University of Pittsburgh Center for Research Computing, which maintains the HTC cluster used for bioinformatic analysis of RNA-seq data. SeqCenter (Pittsburgh, PA) performed whole-genome sequencing. The UPMC Health Sciences Sequencing Center assisted with nucleic acid sample quality control. Jan-Peter van Pijkeren generously donated plasmid pVPL3004. Thank you to Adam Ratner for his thoughtful feedback on a draft version of this manuscript.

## Author contributions

Study conceptualization: T.A.H.; Experimental design and execution: K.P.G., G.H.H., V.H.B., J.L.E., T.A.H.; RNA-seq and transcriptomic analyses: G.H.H., A.D., H.T., T.A.H.; Manuscript preparation: K.P.G., G.H.H., T.A.H.; Editing: K.P.G., G.H.H., V.H.B., A.D., H.T., T.A.H.

## Competing interests

The authors declare no competing interests.
