## [Peer Review File · Communications Biology]

Reviewers' comments:

Reviewer #1 (Remarks to the Author):

In this article, authors want to explore the mechanisms of genome-wide regulatory changes due to Cas9 off-target effects in GBS (Group B streptococcus).

First, based on what was known about SpyCas9 and thanks to the relative proximity of the two species of streptococci, the authors were able to generate a GBS dCas9 to use it as a CRISPRi tool in GBS, making it possible to study the role of essential and non-essential genes in pathophysiology as has been done in other species.

Then, they used different Cas9 variants (dcas9, scas9, Δ cas9) and RNAseq to conduct their study.

Finally, they identify that Cas9 off-target effects in GBS could come from a nonspecific PAM binding.

However, even if they could observe transcriptional changes with their Cas9 variants, these changes did not result in virulence changes in vivo, unlike what was previously observed by other teams.

Major comments:

1. This article is quite long, and it seems that the authors want to convey two rather distinct messages: the construction of the CRISPRi tool in GBS, the demonstration of its functionality and so the possibility to add it to the panel of tools derived from CRISPR-Cas and for understanding GBS physiopathology. On the other side, the work focus on trying to explain the role of the GBS CRISPR-Cas system in the transcriptional regulation, which seems to be the main objective. Even if there is a complementarity between the two parts, authors should consider reducing the first part, or they should even consider submitting two manuscripts if they want to value their great work concerning the development of the CRISPRi tool.

In comparison, the non-scanning mutant (sCas9) is much less detailed despite being the one allowing the elucidation of the genome-wide transcriptional regulation mechanism.

2. It is quite disconcerting that there is no off-target effect observed in CRISPRi experiments (figure 5) although this is a problem often encountered by other teams and above all, that the aim of the authors is to explain the off-target effects of the PAM scanning function of Cas9 on transcriptional regulation (See also lines 447-451 in discussion).

Minor comments:

3. CRISPRi of GBS essential genes: it's interesting that authors observed only a delayed growth phenotype and not a no growth phenotype using a constitutively expressed CRISPRi system (both dCas9 and sgRNA are constitutively expressed). Inducible systems are often preferred for studying essential genes.

4. Figure 5: this figure needs scales for better clarity. Also, there seems to be an inconsistency between the caption and the writings on the figure regarding the data that are represented by black or red histograms.

Lines 227-228: there was no confirmation by qPCR of the RNA seq results on this part?

5. Lines 359-361: confirmatory RT-qPCR experiments are mentioned but the results are not presented or discussed. There is no conclusion: confirmation or not?

6. Lines 423-426: I couldn't see to which part these sentences refer to. Were these results presented?

7. Lines 151-152: the affirmation here does not stand for the two strains, it should be nuanced.

8. Lines 220-224: spacer 886 has not the same effect in the two strains. This result is not discussed.

9. Lines 103, 275, 305, Figure 6 & Supplemental figure 1: there is probably a mistake concerning the location of the PAM Scan2 which must be R1341 and not R1441.

10. Lines 154, 160, 162: nucleotide position is 310 in the text and 316 in figure 2.
11. Line 313: OD value is missing for mid-low growth.
12. Lines 559-560: was it really M17 broth or tryptic soy broth that was used?

Reviewer #2 (Remarks to the Author):

The authors have submitted a very clearly written manuscript describing their experimental findings. They show that chromosomal expression of mutant Cas9 in *Streptococcus agalactiae* strains can be used to inhibit transcription of genes from the bacterial chromosome or from plasmids.

The authors have also used protospacer-adjacent motif (PAM) interaction domain (PID) mutants to expand upon previously published observations that suggested that the CRISPR1 system of GBS regulates endogenous bacterial gene expression. Using RNA seq and differential gene expression analysis, they conclude that PAM binding is implicated as a major contributor to the transcriptional regulatory function of SgaCas9. Through gene set enrichment analysis, the authors identify defense genes and nucleotide metabolism/transport genes as highly down regulated by inhibition of SgaCas9.

Throughout the manuscript experiments are well designed, executed and reported.

Overall, this work adds to the understanding of how the CRISPR1 system contributes to the regulation of genome transcription in GBS – implicating PAM binding as a contributor to genomic transcriptional modulation. Furthermore, this work has shown that a dCas9 mutant expressed from the chromosome can be useful for further studies requiring gene inhibition by introduction of guide RNA to GBS to direct CRISPRi.

Specific comments

1. There is an overall lack of an interpretable phenotype or phenotypes in the modified strains. While it seems clear that there are changes in gene expression profiles – and in particular defense genes and nucleotide metabolism/transport genes – in PID mutants, it is unclear that this is biologically meaningful. This paper would be much more impactful if a biological phenotype related to the altered mRNA profiles could be demonstrated.
2. Although indicated in the set size histograms of figure 7A, this reviewer would find it helpful if the authors would include a comment on the magnitude of the gene expression changes. How many genes were reproducibly up or down regulated in PID mutant v WT for example? And by what degree? Does this agree with previous work of others?
3. In Figure 4A and B there is a difference in cell growth between strains when CcpA 886 is targeted by CRISPRi. Specifically, A909 is unaffected while 10/84 has a significant growth defect. Can the authors explain why this might be?
4. The clarity of Figure 7E would benefit from the inclusion of a unit on the scale for the heat map shown.
5. Throughout the manuscript, the term 'off-target' is used interchangeably with transcription/mRNA expression effects of Cas9. Since off-target has a standardized definition (also defined here) in the realm of genome editing, perhaps the authors could consider using 'transcription effects' or 'mRNA expression' effects instead.
6. Also throughout the manuscript the authors have used 'spyCas9'. However, 'SpyCas9' is the preferred notation to indicate genus and species of the source organism.
7. The term 'sham' throughout the document would be more accurately described as 'non-targeting'.
8. The introduction of SpyCas9 targeting on lines 89-90 contains an error. Strand separation by Cas9

is not ATP-dependent and ATP is not even mentioned anywhere in the indicated reference (34; Ivanov et al. (2020) PNAS).

9. The term 'PAM-scanning', or 'scanning' may imply 1 dimensional target search when current models suggest a more complicated PAM-probing mechanism (see for example Globyte, V., et al. (2019).

Embo J 38. 10.15252/emj.201899466). Please consider an alternative to the term scanning in the manuscript body – perhaps 'probing', 'interaction', 'recognition', 'binding', etc., could be used

10. The discussion - lines 459-467 - would benefit from an expansion on or further description of the results of the other groups who have shown deletion of Cas9 affects virulence in animal models and offer some explanations as to why their results differ from the ones presented here.

11. Consider replacing the word 'matrix' with 'heatmap' on line 326

12. Consider using the term 'total RNA purification' as opposed to 'whole RNA purification' on line 313.

13. P3015b could be a useful tool for other researchers. Will the authors consider depositing it in Addgene for distribution?

We would like to thank our two reviewers for their careful reading and thoughtful critiques of our initial submission. Below are point-by-point responses to the comments we received.

Reviewer #1 (Remarks to the Author):

In this article, authors want to explore the mechanisms of genome-wide regulatory changes due to Cas9 off-target effects in GBS (Group B streptococcus).

First, based on what was known about SpyCas9 and thanks to the relative proximity of the two species of streptococci, the authors were able to generate a GBS dCas9 to use it as a CRISPRi tool in GBS, making it possible to study the role of essential and non-essential genes in pathophysiology as has been done in other species.

Then, they used different Cas9 variants (dCas9, sCas9, Δ cas9) and RNAseq to conduct their study. Finally, they identify that Cas9 off-target effects in GBS could come from a nonspecific PAM binding. However, even if they could observe transcriptional changes with their Cas9 variants, these changes did not result in virulence changes in vivo, unlike what was previously observed by other teams.

Major comments:

1. This article is quite long, and it seems that the authors want to convey two rather distinct messages: the construction of the CRISPRi tool in GBS, the demonstration of its functionality and so the possibility to add it to the panel of tools derived from CRISPR-Cas and for understanding GBS physiopathology. On the other side, the work focus on trying to explain the role of the GBS CRISPR-Cas system in the transcriptional regulation, which seems to be the main objective. Even if there is a complementarity between the two parts, authors should consider reducing the first part, or they should even consider submitting two manuscripts if they want to value their great work concerning the development of the CRISPRi tool.

In comparison, the non-scanning mutant (sCas9) is much less detailed despite being the one allowing the elucidation of the genome-wide transcriptional regulation mechanism.

Thank you for this critique, which we understand. We acknowledge the “two-part” structure of this paper and considered it carefully.

Our primary goal is to describe and share the GBS CRISPRi technology, which we believe will be an accelerant for new research. However, given multiple reports of GBS Cas9 mutants having distinct phenotypes—sometimes significantly affecting virulence characteristics—we thought it was also important to investigate any genome-wide effects of dCas9.

We did not end up finding that dCas9 significantly alters the GBS transcriptome absent a targeting gRNA. At that point, we felt obliged to try to explain why this was the case using the knockout and sCas9 strains. We hope that by presenting the full scope of our inquiry in one article (trying to keep it as contained as possible!) future researchers will feel confident using CRISPRi in pathogenesis models without a cloud of concern that the approach is fundamentally altering GBS virulence at baseline. We hope this manuscript,

once published, will serve as a solid foothold and comprehensive reference for those efforts.

2. It is quite disconcerting that there is no off-target effect observed in CRISPRi experiments (figure 5) although this is a problem often encountered by other teams and above all, that the aim of the authors is to explain the off-target effects of the PAM scanning function of Cas9 on transcriptional regulation (See also lines 447-451 in discussion).

Thank you for this comment (also see Reviewer 2's critique #5 below). We would distinguish here between off-target Cas9 activities in which the enzyme binds one or more target distal from its gRNA complement (perhaps due to partial complementarity), and nonspecific, scanning-related, genome-wide transcriptional changes. In the first case (off-target binding due to gRNA non-specificity), we do not observe this phenomenon with the *cyl* or *covR* targeting protospacers we used here. Note that we use the Broad Institute CRISPick algorithm to select our gRNA sequences, and this algorithm incorporates quality-control processes designed to reduce or eliminate off-target binding. Even so, off-target binding is possible with our Cas9 system and is presumably related to the exact target. For this reason, future users are encouraged to incorporate multiple gRNA sequences and careful validation into their experimental protocols.

The other form of genome-wide transcriptional effects of Cas9 scanning that we describe in the paper are undoubtedly active in the *cyl* and *covR* targeted strains, but those effects would cancel out in the comparison between the targeting and sham-targeted strains diagrammed in Figure 5. To help clarify these two terms, we have replaced "off-target" with "transcription effects" when we are referring to this second kind of genome-wide effect.

Minor comments:

3. CRISPRi of GBS essential genes: it's interesting that authors observed only a delayed growth phenotype and not a no growth phenotype using a constitutively expressed CRISPRi system (both dCas9 and sgRNA are constitutively expressed). Inducible systems are often preferred for studying essential genes.

We also found this growth delay phenomenon curious and wondered about its cause. We are not sure, but we believe an important clue is the fact that the extent of growth delay seems to be a function of target position along the gene, with targets nearer to the start of the coding sequence resulting in more severe growth suppression. This observation argues against stochastic mutations in the plasmid protospacer, *dcas9*, or the gene target itself as an "escape" mechanism, as these events would not be expected to relate to the gRNA target site. We speculate that the delayed growth we observe for knockdowns of these essential genes may reflect gradual accumulation of enough full-length transcript to exceed a threshold needed for growth, and that gRNA targets near the start of a gene may impede this accumulation more than targets nearer the terminus. We added consideration of this important point to our revised Discussion section (lines 479-488).

4. Figure 5: this figure needs scales for better clarity.

Added.

Also, there seems to be an inconsistency between the caption and the writings on the figure regarding the data that are represented by black or red histograms.

Thank you. We corrected this discrepancy.

Lines 227-228: there was no confirmation by qPCR of the RNA seq results on this part?

We did perform RT-qPCR validation of the *cyl* and *covR* knockdowns; these data are presented in Figure 3 along with phenotypic validation data. We did not perform secondary RT-qPCR validation of the CovR regulon changes in the *covR* knockdown strains. This regulon has been well-established and our RNA-seq data aligned closely to what we would expect from those studies.

5. Lines 359-361: confirmatory RT-qPCR experiments are mentioned but the results are not presented or discussed. There is no conclusion: confirmation or not?

We have added a line stating that the RT-qPCR results matched expectations from RNA-seq. Thank you.

6. Lines 423-426: I couldn't see to which part these sentences refer to. Were these results presented?

Apologies—these hypotheses we considered were not fruitful and yielded only negative results. In the interest of parsimony, we are not including output from all those dead ends in this report, but we are happy to share what we tried with anyone who is interested. We have added a clarifying statement to the revised text (line 433).

7. Lines 151-152: the affirmation here does not stand for the two strains, it should be nuanced.

Agreed. This has been addressed in the revised manuscript (see lines 154-156).

8. Lines 220-224: spacer 886 has not the same effect in the two strains. This result is not discussed.

The *ccpA* coding sequence is 1,005 bp long, so the gRNA targeting position 886 is in the terminal 20% of the gene. We believe that in “edge” cases where the RNA polymerase collides with dCas9 at the very end of the gene, a modest amount of full-length transcript may end up being transcribed (an analogy might be a cryptic promoter that inefficiently generates some transcript). CNCTC 10/84 and A909 show very different growth phenotypes across a range of cell stress conditions. We speculate that CcpA requirements may differ between the two strains and that our expression titration by moving the gRNA target identifies experimentally different thresholds for the two strains at which

knockdown leads to major growth delay. We have added this interpretation into the Discussion section of the revised manuscript.

9. Lines 103, 275, 305, Figure 6 & Supplemental figure 1: there is probably a mistake concerning the location of the PAM Scan2 which must be R1341 and not R1441.

Fixed in all locations. We really appreciate that you caught that!

10. Lines 154, 160, 162: nucleotide position is 310 in the text and 316 in figure 2.

Thank you. 316 is the correct number (310 was a carryover from an earlier numbering scheme). This has been corrected in the revised text.

11. Line 313: OD value is missing for mid-low growth.

Thank you for noticing that. The value (0.6) is added.

12. Lines 559-560: was it really M17 broth or tryptic soy broth that was used?

It was M17 for competent cell preparation. This comes from historical protocols and we have never changed them.

Reviewer #2 (Remarks to the Author):

The authors have submitted a very clearly written manuscript describing their experimental findings. They show that chromosomal expression of mutant Cas9 in *Streptococcus agalactiae* strains can be used to inhibit transcription of genes from the bacterial chromosome or from plasmids.

The authors have also used protospacer-adjacent motif (PAM) interaction domain (PID) mutants to expand upon previously published observations that suggested that the CRISPR1 system of GBS regulates endogenous bacterial gene expression. Using RNA seq and differential gene expression analysis, they conclude that PAM binding is implicated as a major contributor to the transcriptional regulatory function of SgaCas9. Through gene set enrichment analysis, the authors identify defense genes and nucleotide metabolism/transport genes as highly down regulated by inhibition of SgaCas9.

Throughout the manuscript experiments are well designed, executed and reported.

Overall, this work adds to the understanding of how the CRISPR1 system contributes to the regulation of genome transcription in GBS – implicating PAM binding as a contributor to genomic transcriptional modulation. Furthermore, this work has shown that a dCas9 mutant expressed from the chromosome can be useful for further studies requiring gene inhibition by introduction of guide RNA to GBS to direct CRISPRi.

Specific comments

1. There is an overall lack of an interpretable phenotype or phenotypes in the modified strains. While it seems clear that there are changes in gene expression profiles – and in particular defense genes and nucleotide metabolism/transport genes - in PID mutants, it is unclear that this is biologically meaningful. This paper would be much more impactful if a biological phenotype related to the altered mRNA profiles could be demonstrated.

Thank you for this and the following thoughtful critiques. We were surprised not to observe differences in major outcomes in the bacteremia model, since we thought that experiment would likely capture phenotypic correlates of the transcriptional changes we observed in our Cas9 variants. Since receiving these reviews, we tried simplifying the experimental system to see if we could tease out measurable phenotypic differences. We used a porcine hemin exposure model that has been employed in prior studies (see Laut, C. L. *et al.* DnaJ and ClpX Are Required for HitRS and HssRS Two-Component System Signaling in *Bacillus anthracis*. *Infect Immun* **90**, e00560-21, 2021) as a controlled way to impose oxidative stress relevant to what bacteria encounter in the bloodstream. Using this *in vitro* model, we found subtle but highly reproducible differences in maximum O.D.₆₀₀, where the WT and *dcas9* strains slightly outperformed the *scas9* and Δ *cas9* strains under challenge conditions. These data are presented in a new Supplemental Figure 9 and are discussed in the Results and Discussion sections of the revised manuscript.

Our interpretation of our results, considering previous studies, is that while Cas9 chromosomal scanning likely imparts genome-wide transcriptional changes, it is not a monolithic effect. Other environmentally responsive signaling pathways function simultaneously and—under the extreme conditions of host invasion—likely dominate so that adaptive responses are still observable. We have tried to articulate this viewpoint in the revised Discussion section.

2. Although indicated in the set size histograms of figure 7A, this reviewer would find it helpful if the authors would include a comment on the magnitude of the gene expression changes. How many genes were reproducibly up or down regulated in PID mutant v WT for example? And by what degree? Does this agree with previous work of others?

Please see the two panels below, which show the RNA-seq gene expression change as a function of locus position on the CNCTC 10/84 chromosome. We find it intriguing that expression changes seem consistently localized to certain chromosomal regions. However, as described in the main text, we were unable to establish a convincing correlate of this phenomenon (e.g. predicted differences in DNA supercoiling, GC-content, etc.).

10/84 Log2 FC OD0.6

10/84 Log2 FC OD1.2

We did compare our differential expression findings to prior studies and did not find a strong correlation among orthologous genes. This suggests to us that factors of DNA local geography are likely important contributors to the effect, since GBS orthologs might have differing neighboring features that alter Cas9 access to the chromosome.

3. In Figure 4A and B there is a difference in cell growth between strains when CcpA 886 is targeted by CRISPRi. Specifically, A909 is unaffected while 10/84 has a significant growth defect. Can the authors explain why this might be?

Thank you. We have added some thoughts on this to lines 489-499 in the revised Discussion section. The *ccpA* coding sequence is 1,005 bp long, so the gRNA targeting position 886 is in the terminal 15% of the gene. We believe that in edge cases where RNA polymerase collides with dCas9 at the very end of a gene, a modest amount of full-length transcript may end up being transcribed (an analogy might be a cryptic promoter that inefficiently generates some transcript). CNCTC 10/84 and A909 show very different growth phenotypes across a range of cell stress conditions. We hypothesize that CcpA requirements may differ between the two strains and that our expression titration by moving the gRNA target identifies experimentally different *ccpA* expression thresholds for the two strains at which knockdown leads to major growth delay. (Also see Reviewer 1 critique #3 on a similar point, answered above and addressed in the revised text lines 479-488).

4. The clarity of Figure 7E would benefit from the inclusion of a unit on the scale for the heat map shown.

Understood. The values here are variance stabilized transcript (VST) counts, which give a relative measure of gene expression levels that are normalized (for sequencing depth) and stabilized (for variance across sample replicates) allowing comparisons between different sample conditions. The values are all relative to each other and internally consistent across the different heatmaps, but the units themselves are arbitrary. We have added a label to the legend and provided further explanation in the Figure 7 caption.

5. Throughout the manuscript, the term 'off-target' is used interchangeably with transcription/mRNA expression effects of Cas9. Since off-target has a standardized definition (also defined here) in the realm of genome editing, perhaps the authors could consider using 'transcription effects' or 'mRNA expression' effects instead.

Agree that this could be clearer (also see Reviewer 1's critique #2, which is similar). To help this, we have changed "off-target" effects to "transcriptional" effects in cases where we mean genome-wide non-specific modulation (i.e. not from gRNA mispairing).

6. Also throughout the manuscript the authors have used 'spyCas9'. However, 'SpyCas9' is the preferred notation to indicate genus and species of the source organism.

Thank you for this edit. Corrected throughout.

7. The term 'sham' throughout the document would be more accurately described as 'non-targeting'.

We would like to keep "sham" if the editor agrees. It's shorter (in an already long article!) and we think it also helps specify that there is a protospacer sequence present in this control conditions (as opposed to an empty cloning site). However, to ensure clarity we have made the point explicit in line 249 that "sham" is our label for an encoded yet non-targeting protospacer.

8. The introduction of SpyCas9 targeting on lines 89-90 contains an error. Strand separation by Cas9 is not ATP-dependent and ATP is not even mentioned anywhere in the indicated reference (34; Ivanov et al. (2020) PNAS).

Thank you for catching/correcting this. We modified the sentence.

9. The term 'PAM-scanning', or 'scanning' may imply 1 dimensional target search when current models suggest a more complicated PAM-probing mechanism (see for example Globyte, V., et al. (2019). Embo J 38. 10.15252/emboj.201899466). Please consider an alternative to the term scanning in the manuscript body - perhaps 'probing', 'interaction', 'recognition', 'binding', etc., could be used.

This is an important point and we have added consideration of one- and three-dimensional PAM scanning to our Discussion section (lines 442-447), as well as this helpful reference. In terms of the word "scanning" versus alternatives, we think it's a semantic matter that—with this added nuance—can reasonably stand as is (we note that the Globyte et al. also uses "scanning" in their text).

10. The discussion - lines 459-467 - would benefit from and expansion on or further description of the results of the other groups who have shown deletion of Cas9 affects virulence in animal models and offer some explanations as to why their results differ from the ones presented here.

We concur and have revised the Discussion section to try to account for differences between our study and others, while (hopefully) not overreaching to explain observations for which we don't have determinative data. Please see lines 462-478, which we hope the reviewer finds responsive to this critique.

11. Consider replacing the word 'matrix' with 'heatmap' on line 326

Replaced. Thank you.

12. Consider using the term 'total RNA purification' as opposed to 'whole RNA purification' on line 313.

Done.

13. P3015b could be a useful tool for other researchers. Will the authors consider depositing it in Addgene for distribution?

We will gladly deposit the plasmid with Addgene. We would be happy to see what other groups are able to do/discover with it.

REVIEWERS' COMMENTS:

Reviewer #1 (Remarks to the Author):

I would like to thank the authors, my concerns have been addressed. I hope they even improved the manuscript. I renew my opinion that this article presents interesting advances for the field.

Reviewer #2 (Remarks to the Author):

Thank you for your thoughtful replies and revisions based on our comments. This is really nice work.